# IDENTIFY DOMINATORS: THE KEY TO IMPROVE LARGE-SCALE MAXIMUM INNER PRODUCT SEARCH

## ABSTRACT

Maximum Inner Product Search (MIPS) is essential for machine learning and information retrieval, particularly in applications that operate on high-dimensional data, such as recommender systems and retrieval-augmented generation (RAG), using inner product or cosine similarity. While numerous techniques have been developed for efficient MIPS, their performance often suffers due to a limited understanding of the geometric properties of Inner Product (IP) space. Many approaches reduce MIPS to Nearest Neighbor Search (NNS) through nonlinear transformations, which rely on strong assumptions and can hinder performance. To address these limitations, we propose a novel approach that directly leverages the geometry of IP space. We focus on a class of special vectors called dominators and introduce the **M**onotonic **R**elative **D**ominator **G**raph (MRDG), an *IP-space-native, sparse, and strong-connected* graph designed for efficient MIPS, offering solid theoretical foundations. To ensure scalability, we further introduce the **A**pproximate **R**elative **D**ominator **G**raph (ARDG), which retains MRDG's benefits while significantly reducing indexing complexity. Extensive experiments on 8 public datasets demonstrate that ARDG achieves a 30% average speedup in search at high precision and reduces index size by $2\times$ compared to state-of-the-art graph-based methods.

## 1 INTRODUCTION

Maximum Inner Product Search (MIPS) is foundational in machine learning and information retrieval (Lewis et al., 2020; Seo et al., 2019), especially with the rise of high-dimensional representations based on inner product or cosine similarity. Applications such as recommendation systems (Xu et al., 2018), query-answering chatbots (Ahmad et al., 2019), multi-modal retrieval (Wang et al., 2024), and Retrieval Augmented Generation (RAG) (Asai et al., 2023) depend on efficiently searching large vector databases to find items that maximize similarity with a query vector. Fast and accurate MIPS leaves more time for complex model inference, thereby enhancing both system performance and user experience.

Despite numerous methods developed for MIPS (Morozov & Babenko, 2018; Guo et al., 2020; Zhao et al., 2023; Guo et al., 2016; Liu et al., 2020), the geometric properties of the inner product space require further exploration. This lack of geometric theoretical support has led most existing approaches (Zhao et al., 2023; Zhou et al., 2019; Shrivastava & Li, 2014) to reduce MIPS to the nearest neighbor search (NNS) problem in a transformed space, allowing established advanced NNS methods to address the MIPS problem. However, this reduction is typically achieved through nonlinear transformations, such as the Möbius Transformation (Zhou et al., 2019) or XBOX Transformation (Zhao et al., 2023). These transformations often impose strong theoretical assumptions, introduce data distortion, and struggle with data updates (Morozov & Babenko, 2018), leading to compromised efficiency and scalability, particularly in high-dimensional settings.

To tackle these challenges, we propose a novel approach that leverages the intrinsic geometry of the IP space without transforming it into a metric space. Our key insight is to identify a set of vectors termed

**self-dominators**, which dominate their respective partitions of space by maintaining a higher inner product with vectors in their subspace than outside ones. Thereby, self-dominators are strong candidates for MIPS solutions when the query resides within their dominating regions. We propose an efficient method to identify self-dominators and subsequently build the **Monotonic Relative Dominator Graph** (MRDG), an *IP-space-native* structure optimized for MIPS. MRDG effectively navigates queries towards self-dominators while enforcing *sparse connections* between these dominators. Despite its sparsity, we prove the reachability and search complexity on the MRDG. Using a common greedy search algorithm (Algorithm 1), the expected length of a query's search path is $\frac{(\rho n)^{1/d} \log \rho n}{d \Delta(\rho n)} + o(1)$, where $\rho$ represents the density of self-dominators and $\Delta$ indicates data density (Fu et al., 2019), and $n$ denotes the number of high-dimensional vectors. Both factors can be considered approximately constant as the data scale $n$ increases.

Although MRDG is theoretically appealing, its indexing complexity of $O(dn^2)$ makes it impractical for large-scale, high-dimensional datasets. To overcome this limitation, we introduce the **Approximate Relative Dominator Graph** (ARDG), which retains the theoretical strengths of MRDG while improving scalability. We achieve this by balancing the locality, connectivity, and sparsity of the graph structure. Specifically, ARDG identifies approximate self-dominators with high accuracy and introduces a hyperparameter to balance the connectivity and sparsity during IP-specialized edge pruning.

Our contributions are as follows: **(1) Deep Exploitation of IP-Naive Geometry**: To the best of our knowledge, we are the first to investigate the specialized geometric properties in IP space for graph-based MIPS, providing strong theoretical guarantees on graph connectivity and MIPS efficiency through the proposed MRDG. **(2) Introduction of** ARDG: a scalable and efficient approximation of MRDG, tailored for large-scale MIPS applications. **(3) Extensive Empirical Validation**: We validate our approach through comprehensive experiments on 8 public datasets varying in cardinality, dimensionality, and modality, demonstrating the theoretical robustness and practical superiority of ARDG. Our method achieves an average **30% speedup** over state-of-the-art techniques at the same high precision, with a **2× reduction** in graph sizes.

## 2 PRELIMINARIES

**Notations.** Let $\mathbb{R}^d$ denote $d$-dimensional real coordinate space. $\{\cdot\}$ is use to denote sets. Let $\mathcal{D} = \{x_1, \dots, x_n\} \subset \mathbb{R}^d$ represent a vector dataset with $n$ points. A query vector is denoted by $q \in \mathbb{R}^d$. $\langle x, y \rangle$ denotes the inner product (IP) between vector $x$ and $y$. $\|x\|$ gives the Euclidean norm of vector $x$, and the Euclidean distance between vectors $x$ and $y$ is denoted as $\|x - y\|$. The Voronoi cell under IP similarity associated with vector $x$ is represented as $V_x$. A graph is defined as $\mathcal{G} = (\mathcal{V}, \mathcal{E})$, where $\mathcal{V}$ is the node set and $\mathcal{E}$ is the edge set. We use $\sup(S)$ to denote the supremum of a set $S$. $O(\cdot)$ and $o(\cdot)$ denote the big $O$ and small $o$ notations, respectively.

**Problem Definition.** The **Maximum Inner Product Search** (MIPS) problem is defined as follows: Given a query vector $q \in \mathbb{R}^d$ and a dataset $\mathcal{D} = \{x_1, x_2, ..., x_n\}$, the goal is to find the vector $x^* \in \mathcal{D}$ that maximizes the inner product with the query: $x^* = \arg\max_{x \in \mathcal{D}} \langle q, x \rangle$.

Similarly, the **Nearest Neighbor Search** (NNS) problem is defined as finding the vector $x^* \in \mathcal{D}$ that is closest to the query $q$ typically under the Euclidean distance: $x^* = \arg\min_{x \in \mathcal{D}} \|q - x\|$.

**Challenges in MIPS and Prior Work.** Although the MIPS and NNS problems appear similar, the approaches to solve each are not directly interchangeable. This is because IP is not a typical metric that accept properties like the triangle inequality, which are fundamental to many NNS algorithms. Methods designed for the NNS problem leverage the triangle inequality to efficiently prune the search space (Malkov & Yashunin, 2018; Tian et al., 2023; Babenko & Lempitsky, 2014; Ram & Gray, 2012; Fu et al., 2019), thereby improving performance. Due to the lack of well-explored geometry favoring search in IP space, most researchers turn to reduce the MIPS problem to an NNS problem and focus on the **approximate** MIPS problem (Zhao et al., 2023; Zhou et al., 2019; Shrivastava & Li, 2014; Yan et al., 2018; Li et al., 2018).

This relaxed problem allows for an acceptable loss in accuracy in exchange for faster query processing or to accommodate transformations into metric spaces. The approximate MIPS problem is defined as follows: Given a query $q \in \mathbb{R}^d$, a dataset $\mathcal{D} \subset \mathbb{R}^d$, and an approximation ratio $\epsilon \in (0, 1)$, let $p^* \in \mathcal{D}$ be the exact MIPS solution for $q$. The goal is to find a vector $p \in \mathcal{D}$ satisfying: $\langle p, q \rangle \geq \epsilon \cdot \langle p^*, q \rangle$.

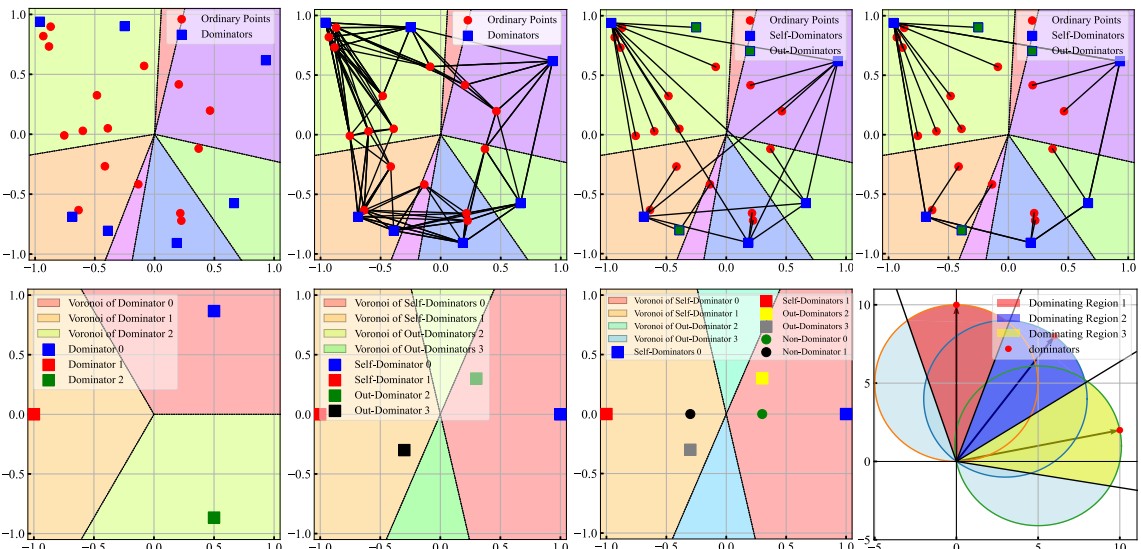

Figure 1: Illustrations of IP geometry concepts, simulated on small toy 2D data. (a) IP-Voronoi cells (open hyper-cones) with associated dominators. (b) ip-NSW Graph. (c) A sparse Naive Dominator Graph (refer to Section 3.1) . (d) MRDG formed through Definition 5. (e) Simpler illustration of self-dominators. (f) Showcase of out-dominators dominating vacant regions. (g) Showcase of ordinary points residing in self-dominators' Voronoi$_{ip}$ cells. (h) Valid dominating region of dominators—capped hyper-cones.

**Transformation-Based Methods.** To address the approximate MIPS problem, several transformations have been proposed to map IP space into Euclidean space, enabling the use of NNS algorithms. Two commonly used transformations are the **Möbius transformation** and the **XBOX transformation**. The Möbius transformation normalizes each vector $p \in \mathbb{R}^d$ to $p/\|p\|^2$. The XBOX transformation applies an asymmetric mapping: query $q$ is mapped to $q' = [q; 0]$, while base vector $p$ is mapped to $p' = [p; \sqrt{M^2 - |p|^2}]$, where $[;]$ denotes vector concatenation. Both transformations introduce non-linear distortions that can affect the data structure (Morozov & Babenko, 2018; Zhao et al., 2023; Zhou et al., 2019). In particular, the XBOX transformation faces challenges with data updates. The hyperparameter $M$ must be chosen carefully based on the norm distribution. A small $M$ cause inflexibility in data updates when the new added vector' norm exceed $M$, while a large $M$ increases distortion by overly influencing the transformed vector norms.

**Graph-Based MIPS.** Graph-based approaches for MIPS have gained attention as a promising research direction. These methods (Morozov & Babenko, 2018; Tan et al., 2021) are primarily developed from the Delaunay graph in IP space, which serves as a dual structure to the Voronoi cell set under the IP similarity. Similar to the Voronoi cells in Euclidean space, the Voronoi cell in IP space is defined as:

**Definition 1** (**IP-Voronoi Cell**). *The Voronoi$_{ip}$ cell $V_x$ associated with a vector $x \in \mathcal{D}$ is defined as:* $V_x = \{y \in \mathbb{R}^d \mid \langle y, x \rangle > \langle y, z \rangle, \forall z \in \mathcal{D}, z \neq x\}$.

**Definition 2** (**Generalized IP-Delaunay Graph**). *Given a dataset $\mathcal{D} \subset \mathbb{R}^d$, the generalized IP-Delaunay Graph $\mathcal{G}$ is constructed by connecting any two nodes $x_i$ and $x_j$ with a bi-directional edge if their corresponding Voronoi$_{ip}$ cells $V_{x_i}$ and $V_{x_j}$ are adjacent in $\mathbb{R}^d$, or $x_i$ is contained in $V_{x_j}$.*

By establishing bidirectional connections within adjacent Voronoi cells, (Morozov & Babenko, 2018) demonstrated that any two nodes in the IP-Delaunay Graph (a subset of generalized IP-Delaunay Graph, we discuss the difference in Appendix C.1) are reachable using a greedy search algorithm (Algorithm 1), which iteratively selects neighbors maximizing IP similarity to query $q$. Although this method is theoretically sound, their proposed ip-NSW algorithm becomes dense in high-dimensional spaces, where each node connects to a large portion of the dataset. Figure 1 illustrates the partitioning of space into Voronoi$_{ip}$ cells (sub-figure (a)) and the corresponding ip-NSW Graph (sub-figure (b)) built on a 2D toy dataset. Vacant Voronoi$_{ip}$ cells may occur depending on point positions.

**Our Motivation.** In this paper, we aim to explore intrinsic geometric properties of IP space benefiting MIPS, akin to how triangle inequality benefits NNS in Euclidean space. We explore a sparse yet efficient graph index for MIPS at scale without relying on transformations that would distort the topology.

---

**Algorithm 1:** GREEDY SEARCH FOR GRAPHS

**Data:** Dataset $\mathcal{D}$, Graph $G$, query $q$, candidate set size $l_s$, result set size $k$. similarity function $s(,)$

**Result:** Top $k$ result set $R$.

$R \leftarrow \emptyset; Q \leftarrow \emptyset;$
$P \leftarrow$ random sample $l_s$ nodes from $G$;
**for** *each node $p$ in $P$* **do**
    $Q.\text{add}(p, s(p, q));$
$Q.\text{make\_max\_heap}(); R.\text{init\_min\_heap}()$
**while** *$Q.size()$* **do**
    $p \leftarrow Q.pop()[0]; R.insert((p, s(p, q)))$
    **if** *visited(p)* **then**
        continue;
    $N_p \leftarrow$ neighbors of $p$ in $G$
    **for** *each node $n$ in $N_p$* **do**
        $Q.insert((n, s(n, q)));$
    $Q.resize(l_s); R.resize(k);$
**return** $R$

---

## 3 THEORETICAL FOUNDATION

In this section, we introduce the concept of dominators under the IP space. By focusing on these dominators, we can leverage the intrinsic geometry of the IP space to improve search efficiency.

### 3.1 DOMINATORS AND CONSTRUCTION OF NAIVE DOMINATOR GRAPH

We begin by defining dominators and exploring its properties in the context of IP space:

**Definition 3** (**Dominator of Voronoi Cell**). *Followed by Definition 1, the vector $x \in \mathcal{D}$ associated with Voronoi$_{ip}$ cell $V_x$ is a **dominator**. $S_{dom}$ represents dominators in $D$.*

**Property 1.** *The Voronoi$_{ip}$ cells associated with the dataset $\mathcal{D}$ provide a finite full space coverage of $\mathbb{R}^d$.*

**Property 2.** *For any query $q \in \mathbb{R}^d$, its MIPS solution lies within the dominator set $\mathcal{S}_{dom} \subset \mathcal{D}$.*

From Property 1 and 2, we conclude that executing MIPS on the entire dataset $\mathcal{D}$ can be reduced to MIPS on the dominator set $\mathcal{S}_{dom}$. This insight motivates the construction of a graph structure that focuses mainly on dominators, given the success of prior graph-based methods (Malkov & Yashunin, 2018; Fu et al., 2019).

**Definition 4** (**Naive Dominator Graph (NDG)**). *Given a dataset $\mathcal{D} \subset \mathbb{R}^d$, a **Naive Dominator Graph (NDG)** $\mathcal{G}$ can be constructed as follows: For each point $x_i \in \mathcal{D}$, sort the remaining points in descending order of $\langle x_i, x_j \rangle$ to form a list $L(x_i)$. Starting from the beginning of $L(x_i)$, evaluate each point $x_j$ using*

*these conditions: **(1):** $\langle x_j, x_j \rangle \geq \langle x_j, x_k \rangle$ for all $k < j$. **(2)** $\langle x_k, x_k \rangle \geq \langle x_j, x_k \rangle$ for all $1 < k < j$. If $x_j$ satisfies these conditions, add a bi-directional edge $(x_i, x_j)$ to the graph $\mathcal{G}$.*

**Theorem 1.** *Given a dataset $\mathcal{D} \subset \mathbb{R}^d$, a Naive Dominator Graph (NDG) is: (1) a strongly connected graph; (2) able to identify $\mathcal{S}_{dom}$; and (3) containing the fully connected graph of $\mathcal{S}_{dom}$.*

In the proof of Theorem 1 (Appendix B.2), we further distinguish between two types of dominators based on their relationship to their Voronoi$_{ip}$ cells: A **self-dominator** resides within its own Voronoi$_{ip}$ cell, meaning if for all $y \in \mathcal{D}$, $\langle x, x \rangle > \langle x, y \rangle$, then $x$ dominates itself and belongs to $V_x$. An **out-dominator** does not reside in its own Voronoi$_{ip}$ cell, meaning that $\exists y \in \mathcal{D}$ such that $\langle x, x \rangle \leq \langle x, y \rangle$, then $x$ is potentially dominated by $y$. Theorem 1 identifies dominators and builds a graph linking nodes to these dominators. Because MIPS solutions reside within the dominator set (Property 2), we can prune the graph by keeping only one self-dominator neighbor for each ordinary point, resulting in a sparser NDG (Figure 1 (c)).

Moreover, Sub-figures (e) to (h) in Figure 1 illustrate self-dominators, out-dominators, and ordinary points. We can see that the Voronoi$_{ip}$ boundaries are determined by the dominators, while ordinary points do not contribute to the formation of boundaries. Additionally, a self-dominator dominates a capped hyper-cone (sub-figure (h)) according to the definition. The cap is part of the surface of a hypersphere centered at $x/2$ with diameter $\|x\|$, due to the condition $\langle x, x \rangle > \langle y, x \rangle$, where $y$ is any point potentially dominated by $x$. The valid dominating region contains at least one point—the self-dominator itself—while out-dominators do not reside in their own dominating regions, potentially leading to vacant dominating regions.

With above observations, we are more interested in **self-dominators** with respect to MIPS for three reasons:

**(1)** Self-dominators tend to have large norms, as their squared norms exceed the IP values with any other vectors. Prior studies (Liu et al., 2020; Tan et al., 2019) have also shown MIPS solutions often cluster around large-norm points theoretically and empirically, making self-dominators strong MIPS candidates. **(2)** Using the edge selection from Definition 4, identifying self-dominators is straightforward, ensuring every point is linked to at least one, ensuring efficient graph traversal via self-dominators. **(3)** Ordinary points in out-dominators' Voronoi$_{ip}$ are not MIPS solutions, while out-dominators likely belong to the Voronoi$_{ip}$ of self-dominators. This implies MIPS solutions are concentrated within self-dominators's Voronoi$_{ip}$ cells.

### 3.2 IMPROVE MIPS EFFICIENCY WITH MONOTONIC RELATIVE DOMINATOR GRAPH

It is practical to assume that queries primarily fall within the union of Voronoi$_{ip}$ cells associated with self-dominators. In modern machine learning, query vectors are often trained to align with the distribution of base vectors (Bengio et al., 2013; Chen et al., 2020), making them more likely to be near self-dominators, which dominate larger regions. For these queries, MIPS can be streamlined by focusing solely on self-dominators. Thus, the search efficiency on a graph built around self-dominators depends on two factors: the density of self-dominators in the dataset and the efficiency of navigating through them.

To begin, we can estimate the ratio of self-dominators in a dataset as follows:

**Theorem 2.** *Given a dataset $\mathcal{D} \subset \mathbb{R}^d$ where vectors are element-wise i.i.d. and drawn from the standard Gaussian distribution $\mathcal{N}(0, 1)$, the probability that a vector $x \in \mathcal{D}$ with norm $\|x\| = r$ is a self-dominator is $\mathcal{P}_{dom}(x) = \Phi(r)$, where $\Phi(\cdot)$ is the cumulative distribution function (CDF) of the standard Gaussian.*

Theorem 2 also confirms that self-dominators are predominantly among vectors with large norms. Surprisingly, we can infer from Theorem 2 that if a vector $x$ has a norm larger than 4, it is almost certain to be a self-dominator. The proportion of such vectors is closely related to the data distribution and the dimensionality $d$. Since the distribution of $\|x\|$ follows a chi distribution when the elements of $x$ are i.i.d. samples from $\mathcal{N}(0, 1)$, the number of self-dominators can be estimated by: $nP(\|x\| > r) = n\left(1 - \frac{\gamma(d/2, r^2/2)}{\Gamma(d/2)}\right)$, where $\gamma(\cdot)$ is the lower incomplete gamma function and $\Gamma(\cdot)$ is the gamma function.

Via numerical integration, we can estimate the self-dominator density from the given formula. A visualization of $P(\|x\| > r)$ is provided in Appendix: In high-dimensional spaces, nearly all vectors become self-dominators, reflecting the curse of dimensionality. However, this is less problematic for real-world datasets for two reasons: **(1)** Real-world data typically lies on low dimensional manifolds (in estimated dimension $d'$) with structured, correlated dimensions, making the search difficulty similar to that of i.i.d. datasets with dimension $d'$ (Fu et al., 2019). **(2)** The norms of real-world vectors are usually regularized or bounded, unlike i.i.d. Gaussian distributions. In our experiments, the 300-$d$ Netflix and 784-$d$ MNIST1M datasets contain only 3.2% and 6.2% self-dominators respectively, significantly reducing the search space.

Although focusing on self-dominators reduces the candidate set, searching on the graph $\mathcal{G}$ from Definition 4 is still inefficient due to its dense connectivity among self-dominators. To address this, we further prune $\mathcal{G}$ using the strategy from Monotonic Relative Neighborhood Graph (MRNG) (Fu et al., 2019):

**Definition 5** (**MRDG**). *Given a dominator graph $\mathcal{G}$ defined on $\mathcal{D} \subset \mathbb{R}^d$ with dominators identified and connected using Definition 4, we construct the **Monotonic Relative Dominator Graph (**MRDG**)** $\mathcal{G}^*$ by further pruning each node's self-dominator neighbors as follows: (1) Sort the self-dominator neighbors of each node $x$ by their Euclidean distance to $x$ in ascending order to form a list $L = \{y_i\}$. (2) For each $y_{i+1}$ in $L$, prune it if there exists $y_j$ with $1 \leq j \leq i$ such that $\|y_{i+1} - x\| > \|y_{i+1} - y_j\|$.*

Definition 5 can also be viewed as the construction algorithm of MRDG. The strategy used in the definition efficiently sparsifies the graph while maintaining short paths between nodes, a technique widely used in Euclidean space for NNS methods like HNSW (Malkov & Yashunin, 2018) and NSG (Fu et al., 2019). Sub-figure (d) in Figure 1 illustrates the proposed MRDG, which is even sparser than the NDG shown in sub-figure (c) while still maintaining the necessary connections to fulfill efficient MIPS.

### 3.3 EFFICIENCY OF EXECUTING MIPS ON MRDG

We now discuss how efficiently MIPS can be performed on an MRDG. A key question arises: How does pruning based on the Euclidean metric improve search performance under IP similarity? We demonstrate that for any query $q$ whose MIPS solution is a self-dominator, it can be efficiently retrieved with the widely used greedy search Algorithm 1 on the MRDG $\mathcal{G}^*$.

**Theorem 3.** *Given a dataset $\mathcal{D} \subset \mathbb{R}^d$ and an MRDG $\mathcal{G}^*$ defined on $\mathcal{D}$, the MIPS solution among self-dominators for any query $q$ is reachable via Algorithm 1 starting from any node $p$.*

Theorem 3 shows that the proposed MRDG ensures good connectivity and effective coordination with Algorithm 1, inheriting these properties from the MRNG (Fu et al., 2019). Consequently, we can derive the expected search path length for MIPS on an MRDG based on the theoretical results of MRNG as follows:

**Theorem 4.** *Given a dataset $\mathcal{D} \subset \mathbb{R}^d$ with $n$ points and an MRDG $\mathcal{G}^*$ defined on $\mathcal{D}$, the expected search path length for MIPS using Algorithm 1 is $\mathbb{E}[L_{path}] = \frac{c(\rho n)^{1/d} \log(\rho n)}{d\Delta(\rho n)} + o(1)$ where $c$ absorbs all constants, $\rho$ is the density of self-dominators in $\mathcal{D}$, and $\Delta(n)$ is a very slowly decreasing function of $n$ (Fu et al., 2019).*

From Theorem 4, the search path length grows at a rate close to $O(\log n)$, enabling fast traversal of the graph. This allows us to estimate the search complexity of MIPS on an MRDG. According to (Fu et al., 2019), the maximum out-degree of an MRNG is bounded by a constant $R$ related to the dimensionality $d$ and independent of $n$, so the maximum out-degree of an MRDG is similarly capped. Thus, the search time complexity on an MRDG is approximately $O(\frac{Rc(\rho n)^{1/d} \log \rho n}{d\Delta(\rho n)})$. The growth rate is dominated by $(\rho n)^{1/d} \log \rho n$, , which approaches $\log \rho n$ for small $\rho$, ensuring efficient amortized search performance, especially when $\rho$ is small.

## 4 PRACTICAL APPROXIMATION: ARDG

While MRDG is efficient for MIPS in theory, constructing it for large-scale datasets is impractical due to its $O(dn^2)$ indexing complexity. To address this, we propose the **Approximate Relative Dominator Graph** (ARDG), designed to reduce indexing costs. ARDG focuses on **Locality**, **Connectivity**, and **Sparsity**.

### 4.1 INDEXING METHODOLOGY

**Locality** is a key principle in proximity graphs for vector-based search under common metrics (Wang et al., 2021; Jayaram Subramanya et al., 2019). In top-$k$ MIPS, maintaining locality—by connecting neighbors with the largest IP similarity—enhances retrieval efficiency: If $x$ is the top-1 solution for a query, the remaining $k - 1$ solutions are likely in $x$'s local neighborhood, allowing the algorithm to focus on the most relevant areas. To preserve locality and speedup indexing, we use an efficient algorithm to build an approximate $k$-MIP graph as the initial structure. Specifically, we use ScaNN to approximately retrieve $k$ neighbors with the largest IP similarity for each point. These $k$-MIP neighbors are then used as candidates in further steps, ensuring good locality in the final ARDG, in line with the requirement of Theorem 1.

**Connectivity.** As per Theorem 1 and the analysis in Section 3.3, it is essential to connect each node to at least one self-dominator and maintain connectivity among self-dominators. To accurately identify self-dominators, we apply dominator-oriented edge selection over a broader range of neighbors. Since building a large $k$-MIP graph is costly, we first construct a $k$-MIP graph with a small $k \ll n$ and enrich the candidates by including 2-hop neighbors. Self-dominators are then selected from the 2-hop neighbors via Theorem 1.

**Sparsity.** To approximate MRDG, we sparsify the intermediate graph using the edge selection strategy in Definition 5. However, we empirically find this can reduce MIPS efficiency by weakening connectivity to self-dominators for two reasons: **(1)** Self-dominators approximately identified in earlier steps, called **Pseudo Self-Dominators**, are selected from limited neighborhood area and may be dominated by others. **(2)** The edge selection strategy in Definition 5 may prune large-norm self-dominators yet retain small-norm ones due to the triangle inequality. Executed on limited candidates, it may harm connectivity between non-self-dominators and self-dominators. To address this, we introduce a parameter $\alpha \in (0, 1)$ that controls the fraction of pseudo dominators being further sparsified. Specifically, $\alpha$ proportion of the pseudo self-dominators closest to $x$ in IP metric will be retained in the final neighbors of $x$, while the rest will be further sparsified, thereby balancing locality, connectivity, and sparsity.

---

**Algorithm 2:** ARDG Indexing

**Data:** Dataset $\mathcal{D}$, max out-degree $R$, integer $k$, ratio $\alpha$

**Result:** ARDG: $\mathcal{G}$

$\mathcal{G} \leftarrow$ Build $k$-MIP Graph with ScaNN;

**for** *each point $p$ in $\mathcal{D}$* **do**
    $N_p \leftarrow$ 2-hop neighbors from $\mathcal{G}$;
    $E_1 \leftarrow$ filter $N_p$ with Theorem 1;
    $E_2 \leftarrow$ filter $N_p$ with Definition 5;
    $E_2 \leftarrow E_1 \backslash E_2$ ▷ remove duplicates
    $E_1$.sort(); ▷ descending in IP dist
    $E_2$.sort(); ▷ ascending in $L_2$ dist
    $E_1$.resize($\alpha R$);
    $E_2$.resize(($1 - \alpha$)$R$);
    $\mathcal{G} \leftarrow \mathcal{G} \cup (E_1 \cup E_2)$
**end**
return $\mathcal{G}$;

---

Empirically, sparsifying the rest $1 - \alpha$ pseudo self-dominators has negligible impact on search efficiency compared to directly sparsifying the $(1 - \alpha)$ 2-hop $k$-MIP neighbors. This is because: **(1)** Both methods produce similar filtered neighbor sets. **(2)** Directly sparsifying the 2-hop $k$-MIP neighbors does not harm connectivity and may potentially enhance locality by introducing additional relevant edges. Therefore, we simplify the process by directly sparsifying the 2-hop $k$-MIP neighbors to fill the $1 - \alpha$ proportion of nodes in the final neighbor set. Finally, we limited the max out-degree of ARDG to $R$ for efficient memory utilizing.

The complete indexing algorithm of ARDG is detailed in Algorithm 2.

## 4.2 COMPLEXITY ANALYSIS

**Indexing Complexity.** ARDG indexing involves two stages: constructing the approximate $k$-MIP graph and sparsifying it. The $k$-MIP graph construction, using ScaNN, has an empirical complexity of $O(nd \log n)$ (Guo et al., 2020), where $n$ is the number of data points and $d$ is the dimensionality. Since we set a maximum degree limit for each point, we can assume each node's 2-hop neighbors is capped by a constant $r$, the edge selection strategy incurs a complexity of $O(r^2 d)$ per node, leading to $O(nr^2 d)$ for the dataset. Thus, the total indexing complexity is $O(nd \log n + cndr^2)$, where $c$ is a constant. The $O(nd \log n)$ term dominates the growth rate as $n$ increases given $r \ll n$.

**Search Complexity.** As ARDG approximates MRDG, its search complexity is analytically intractable due to potential effects on dominator accuracy and path length. However, empirical evaluations show that the search complexity for top-$k$ MIPS solutions on ARDG is only slightly higher than $O(d \log n)$, demonstrating that ARDG closely approximates MRDG while maintaining scalability.

**Worst Case Analysis** can be found in Appendix.

## 5 EXPERIMENTS

In this section, we conduct empirical evaluations of our theoretical findings and the proposed ARDG method. We focus on the following research questions: **RQ1**: How does ARDG perform in terms of search and indexing compared to state-of-the-art methods? **RQ2**: How does the self-dominator ratio affect ARDG's performance? **RQ3**: How scalable is ARDG for large-scale applications? **RQ4**: How sensitive is ARDG to parameter variations in indexing and search?

## 5.1 EXPERIMENT SETUP

**Datasets.** We use eight real-world datasets with varying cardinality, dimensionality, and modality to comprehensively evaluate ARDG's performance. The datasets include the following: **Netflix**, **YahooMusic**, and **Music100**, which are publicly available datasets collected from large-scale online recommender systems, with vectors generated via matrix factorization; **UKBench** and **Deep10M**, widely used public image search datasets, where the vectors encode images into feature space using deep learning models; and **MNIST**, a well-known

Table 1: Dataset Statistics. Ratio represents self dominator ratio.

| Dataset | Base Size | Query Size | Dim | Ratio |
|---|---|---|---|---|
| Netflix | 17,770 | 1,000 | 300 | 3.2% |
| MNIST | 60,000 | 10,000 | 784 | 5.4% |
| YahooMuisc | 136,736 | 1,000 | 300 | 2% |
| Music100 | 1,000,000 | 10,000 | 100 | 29.3% |
| Text2Image1M | 1,000,000 | 100,000 | 200 | 99% |
| MNIST1M | 1,000,000 | 10,000 | 784 | 6.2% |
| UKBench | 1,097,907 | 1,000 | 128 | 97.5% |
| Deep10M | 10,000,000 | 10,000 | 96 | 98% |

dataset for handwritten digit recognition. We directly use the flattened grayscale images as vectors since the flattened format is sufficiently discriminative, and we treat the data as a commonly used representation: multi-hot vectors. **MNIST1M** is a large-scale version of MNIST, where new images are generated through data augmentation (Tan et al., 2021). Finally, **Text2Image1M** is a cross-modal dataset where the query vectors encode text and the base vectors encode images using a jointly trained deep learning model. More details can be found in Table 1.

**Baselines.** We compare ARDG against several recent advanced methods of varying types. **Fargo** (Zhao et al., 2023) is a state-of-the-art LSH method. **ScaNN** (Guo et al., 2020) is a quantization method that integrates the recent state-of-the-art method **SOAR** (Sun et al., 2024) to further enhance performance. **Möbius Graph** (Zhou et al., 2019) is a graph-based method that reduces MIPS to NNS using the Möbius transformation. **ip-NSW** (Morozov & Babenko, 2018), **ip-NSW+** (Liu et al., 2020), **IPDG** (Tan et al., 2019), and **NAPG** (Tan et al., 2021) are famous graph-based methods for MIPS in IP-native space.

**Implementation.** We use the public implementation of the *ScaNN* library, written in C, while all other baselines are implemented in C++. All experiments are conducted on the same machine, with 48 threads used to build the indices for all methods. For query execution, we disable additional compiler optimizations that unrelated to the algorithm and use same number of threads to ensure a fair comparison. Each experiment is run three times, and the average is reported to minimize system variability. For graph based algorithms, we apply a unified parameter configuration across all datasets, focusing on achieving strong average performance. For Fargo and ScaNN, we use the recommended parameters from their papers based on the size of dataset. This setting favors the methods with low parameter sensitivity, which is a valuable feature for users who may not be familiar with the intricacies of algorithm tuning. **Our codes are available in Supplementary Materials**.

**Evaluation Protocol.** An effective MIPS method should provide fast and accurate query processing while minimizing indexing resources to allow for efficient system updates. We evaluate the query performance using the metric **Recall vs. Queries Per Second (QPS)**, which represents the number of queries an algorithm can process per second at each specified $recall@k$ level. The recall@k is formally defined as: $recall@k = \frac{|R \cap R'|}{|R|} = \frac{|R \cap R'|}{k}$, where $R$ is the truth set of results, and $R'$ is the set of results returned by the algorithm. We use $k = 100$ in this paper. The index size and indexing time are reported to evaluate indexing costs.

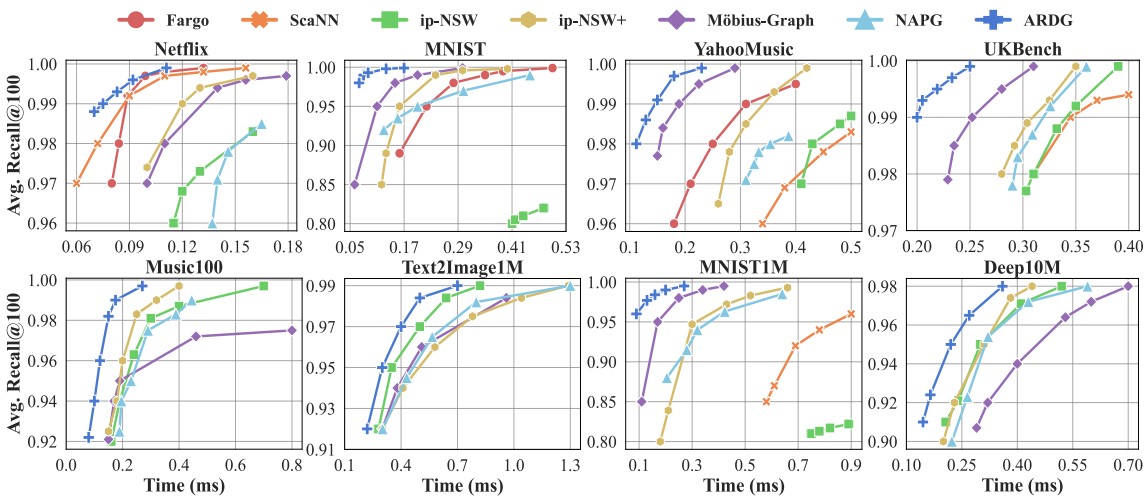

Figure 2: Experimental results of query process performance. Upper left is better.

## 5.2 RESULTS

**Query Process. (RQ1 and RQ2)** Figure 2 presents the query processing performance of ARDG compared to the baselines. Fargo and ScaNN (with SOAR) are absent from some figures (e.g., Deep10M) due to the noticeable gap with graph algorithms on large-scale data. The key findings are as follows: **(1)** ARDG consistently outperforms all baselines across all datasets, owing to its strong theoretical foundations. As a good approximation of MRDG—a sparse, MIPS-oriented graph—ARDG delivers high search efficiency with solid theoretical guarantees. **(2)** ARDG demonstrates robust performance across varying cardinality, dimensionality, and modality, which further reinforces the validity of our theoretical findings. This robustness makes ARDG a versatile method for a wide range of real-world applications. **(3)** For RQ2, ARDG achieves particularly large improvements over other methods on datasets with a lower self-dominator ratio ($\rho$). For example, it outperforms Möbius Graph on MNIST1M ($\rho = 6.2$) with a speedup of nearly 50% at recall@100 = 0.99. On datasets like Text2Image1M ($\rho = 0.99$), the speedup is smaller but still significant at

17% over second-best ip-NSW with a recall of 0.99, which aligns with our theoretical analysis in Section 3. **(4)** Even on datasets with small $\rho$, ARDG continues to outperform the baselines by a moderate margin. This is because ARDG sparsifies connections among self-dominators while maintaining greedy-reachability, as outlined in Definition 5 and Theorem 3. **(5)** IPDG lags behind all graph-based methods and is absent from the figures due to inefficient edge selection among random neighbors, leading to too many isolated sub-graphs, unlike ARDG, which maintains global connectivity (see Appendix D.7). **(6)** Lastly, all other graph-based baselines, except ARDG, encounter precision bottlenecks on certain datasets. This happens because ARDG emphasizes global connectivity in its indexing process, whereas others may suffer from multiple isolated sub-graphs within their graphs.

**Indexing (RQ1).** Appendix D.2 summarizes the indexing statistics. Fargo and ScaNN are the fastest in indexing, with $10\times$ to $100\times$ speedup over graph-based methods due to their simpler algorithms and near $O(nd)$ growth rate. However, their search performance lags far behind, especially on large-scale datasets, where search speed is critical for user experience. Among graph-based methods, Möbius Graph and ARDG are the fastest, with similar indexing times, while ip-NSW+ is the slowest. Fargo has the smallest index size, thanks to its compact hashing tables. Although ScaNN is a quantization method, its index size is comparable to graph-based methods due to the additional structures needed for efficient retrieval. ARDG has the smallest index size among graph-based methods, roughly $2\times$ smaller than others, due to its sparsity. Appendix D.3 demonstrates that ARDG achieves can index a dataset as large as Deep10M (10 million data points) in approximately one hour. These results underscore ARDG's memory efficiency and cost-effectiveness, offering superior search performance while optimizing space usage.

**Scalability (RQ3).** Appendix D.3 shows ARDG exhibits a near $O(\log n)$ growth rate for top-1 MIPS, and between $O(\log n)$ and $O(n^{1/20} \log n)$ for top-100 MIPS. This aligns with the analysis in Section 3.3 and 4.2, further confirming ARDG as an effective approximation of MRDG.

**Parameter Sensitivity (RQ4).** In our empirical study (see Appendix D.5), we examine the relationship between ARDG's search performance and key indexing parameters: $k$ (for the $k$-MIP graph), max out-degree $R$, and balance factor $\alpha$. We find that increasing $k$ improves search performance due to broader neighborhood coverage, but the gains quickly plateau as a small $k = 200$ is sufficient to identify enough self-dominators for connectivity. The performance curve for $R$ is concave, with an optimal value that is easy to determine, though the performance is not highly sensitive to $R$. The parameter $\alpha$, which balances locality and connectivity, affects search efficiency differently: lower $\alpha$ improves top-1 MIPS but reduces top-100 MIPS efficiency. This is because lower $\alpha$ emphasizes transitions among self-dominators, reducing local neighborhood connectivity, confirming that top-1 MIPS solutions mostly fall within the self-dominator set, while top-100 MIPS solutions may include out-dominators and ordinary points (Section 3). We recommend using $k = 200$, $R = 48$, and $\alpha = 0.5$ by default for its robust performance across eight diverse datasets.

**Detailed Data and More Experimental Evaluations** are given in the Appendix due to space limitation.

## 6    CONCLUSION

This paper investigate the geometric properties that enhance similarity search in inner product space, establishing robust theoretical foundations. Our framework deepens the understanding of the topology underlying this widely used similarity measure, with implications that extend beyond information retrieval. To facilitate practical applications, we introduce ARDG, a novel graph-based MIPS method that approximates its theoretical counterpart, MRDG. ARDG achieves an optimal balance between locality, connectivity, and sparsity, consistently outperforming state-of-the-art methods across 8 real-world datasets. Extensive experiments demonstrate the efficiency, robustness, and scalability of ARDG, achieving a remarkable 30% average speedup over the second-best competitor, a $2\times$ reduction in graph size, and moderate indexing time. Additionally, our experimental results provide valuable insights into the structure of inner product space.

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

# Part I

# Appendix

## Table of Contents

## A    RELATED WORKS

**Inner Product** is crucial in AI and machine learning applications such as representation learning (Wang et al., 2024), classification (Yu et al., 2014), clustering (Fujiwara et al., 2023), language modeling (Asai et al., 2023), knowledge graphs (Xu et al., 2020), computer vision (Radford et al., 2021), and recommender systems (Huang et al., 2020). Efficient MIPS is particularly essential for reducing system latency and improving user experience in emerging RAG applications. MIPS methods are generally categorized into Locality Sensitive Hashing (LSH), tree-, quantization-, and graph-based approaches:

**LSH-based methods**: Traditional LSH, originally designed for Euclidean space, is adapted for MIPS using transformations such as $L_2$ (Shrivastava & Li, 2014), Correlation (Shrivastava & Li, 2015), and XBOX (Bachrach et al., 2014). Range-LSH (Yan et al., 2018) is the first to observe that MIPS results cluster around large-norm vectors. Simple-LSH (Neyshabur & Srebro, 2015) introduce a symmetric LSH that enjoys strong guarantees. Fargo (Zhao et al., 2023) represents the recent state-of-the-art. **Tree-based methods**: Early MIPS approaches favored tree structures but struggled with high dimensionality. ProMIPS (Song et al., 2021) addresses this by projecting vectors into a lower-dimensional space, though information loss remains a challenge. LRUS-CoverTree (Ma et al., 2024) improves on this but faces difficulties with negative inner product values. **Quantization-based methods**: NEQ (Dai et al., 2020) quantizes the norms of items in a dataset explicitly to reduce errors in norm. ScaNN (Guo et al., 2020) integrates "VQ-PQ" with anisotropic quantization loss, while SOAR (Sun et al., 2024) employs an orthogonality-amplified residual loss, achieving state-of-the-art performance and is integrated into the ScaNN library. **Graph-based methods**: Proven effective for NNS, graph-based methods have been adapted for MIPS. ip-NSW (Morozov & Babenko, 2018) first uses IP-Voronoi cell and constructs approximate Delaunay graphs using the inner product similarity. ip-NSW+ improve its graph quality by adding an angular proximity graph, while Möbius-Graph employs a Möbius transformation to perform greedy search on a transformed graph. IPDG (Tan et al., 2019) focuses on identifying extreme points and uses a heuristic pruning strategy for top-1 MIPS. NAPG (Tan et al., 2021) introduces a non-continuous inner product-based metric, $\alpha\langle x, y \rangle$, with $\alpha$ varying based on the norm of neighbors. They modify ip-NSW using this new metric and claim state-of-the-art performance among graph-based methods.

**Comparison between IPDG and our work.** IPDG (Tan et al., 2019) defines extreme points as those dominating non-empty Voronoi$_{ip}$ cells, a concept intermediate between the out-dominator and self-dominator in our framework. In our definition (Section 3), Voronoi$_{ip}$ cells dominated by self-dominators are never empty, whereas out-dominators may dominate empty Voronoi$_{ip}$ cells, as demonstrated in Figure 1. IPDG uses a heuristic edge pruning strategy where a neighbor $x_j$ is retained if $\langle x_j, x_j \rangle > \langle x_i, x_j \rangle$ for all neighbor $x_i$ inserted before $x_j$. This differs from our strategy (Theorem 1), which also requires that $\langle x_i, x_i \rangle > \langle x_i, x_j \rangle$ for all $x_i$, providing stronger guarantees. From our perspective, IPDG's approach may incorrectly link out-dominators and even ordinary points as neighbors, resulting in a dense graph that contradicts their goal of connecting only "extreme points". Furthermore, IPDG does not provide a theoretical analysis for this heuristic. In contrast, ARDG inherits the theoretical guarantees of MRDG by approximating it, not only through the edge pruning strategy in Theorem 1 to identify self-dominators, but also by sparsifying connections among self-dominators using Definition 5, resulting in much sparser graphs with solid theoretical foundations.

## B    PROOF OF THEOREMS

To avoid ambiguity when $\langle y, x_i \rangle = \langle y, x_j \rangle$, we assign $y$ to the Voronoi$_{ip}$ cell of $x_i$ if $i < j$. This convention is omitted in all later proofs for simplicity.

## B.1 PROPERTIES OF DOMINATORS

*Proof.* For any query $q \in \mathbb{R}^d$, there exists some $x_i \in \mathcal{D}$ such that $\langle q, x_i \rangle$ is maximized, i.e., $\langle q, x_i \rangle > \langle q, x_j \rangle$ for all $x_j \in \mathcal{D}$ with $j \neq i$. By Definition 3, $x_i$ is the dominator of its Voronoi$_{ip}$ cell $V x_i$ and is also the MIPS solution for $q$. Thus, Proposition 2 holds.

Since the inner product $\langle q, x_i \rangle$ is a continuous function and each $x_i$ is fixed, every point $q \in \mathbb{R}^d$ belongs to at least one Voronoi$_{ip}$ cell $V x_i$. Therefore, the Voronoi$_{ip}$ cells associated with $\mathcal{D}$ fully cover $\mathbb{R}^d$. Given that $\mathcal{D}$ is finite, the set of Voronoi$_{ip}$ cells is also finite, proving Proposition 1. □

## B.2 PROOF OF THEOREM 1

*Proof.* Consider a point $x_i \in \mathcal{D}$ and the sorted list $L(x_i) = [y_1, y_2, \ldots, y_m]$. By definition, we have $\langle y_1, x_i \rangle > \langle y_j, x_i \rangle$ for all $j$ with $j > 1$. Therefore, $x_i$ lies within the Voronoi$_{ip}$ cell dominated by $y_1$, making $x_i$ directly connected to $y_1$, the dominator of this cell.

For any remaining point $y_j$ satisfying the two conditions in $L(x_i) \setminus y_1$, the conditions ensure that:

1. For any pair of $y_j$ and $y_k$ with $k < j$, we have $\langle y_j, y_j \rangle \geq \langle y_j, y_k \rangle$.

2. For any pair of $y_j$ and $y_l$ with $l > j$, we have $\langle y_j, y_j \rangle \geq \langle y_j, y_l \rangle$.

This confirms that the point $z$ maximizing $\langle z, y_j \rangle$, for $z \in \mathcal{D}$, is $y_j$ itself. Hence, $y_j$ does not belong to any other Voronoi$_{ip}$ cell associated with any $z \neq y_j$, affirming $y_j$ as a dominator, dominating at least itself.

In conclusion, base on these rules, each point $x_i \in \mathcal{D}$ is linked to a closest dominator $y_1$ which dominates at least one node $x_i$, and meanwhile $x_i$ is connected to all dominators which at least dominating itself, ensuring that all points are connected to at least one dominator, and dominators are interconnected via the dominators which dominates themselves. Furthermore, since each node is connected to a dominator, and dominators form a fully connected graph, we can conclude that **NDG** is a strongly connected graph. □

## B.3 PROOF OF THEOREM 2

*Proof.* By definition, $x$ of norm $r$ is a self-dominator with the probability:

$$\mathcal{P}_{dom}(x) = P(\langle x, x \rangle > \langle x, y \rangle \mid \|x\| = r) \tag{1}$$

Given $\|x\| = r$, $\langle x, x \rangle = r^2$ is deterministic. Since $y$ is independent of $x$ and its elements are i.i.d., conditioned on $\|x\| = r$, the inner product $\langle x, y \rangle$ becomes a linear combination of $d$ independent standard Gaussian and follows:

$$\langle x, y \rangle \mid \|x\| = r \sim \mathcal{N}(0, r^2) \tag{2}$$

By plugging in and rearranging, equation (1) becomes:

$$\mathcal{P}_{dom}(x) = P\left(\frac{\langle x, y \rangle}{r} < r \mid \|x\| = r\right) \tag{3}$$

Since by standardizing, $Z = \frac{\langle x, y \rangle}{r} \sim \mathcal{N}(0, 1)$, we have:

$$\mathcal{P}_{dom}(x) = P(Z < r) = \Phi(r) \tag{4}$$

where $\Phi(\cdot)$ is the CDF of the standard Gaussian distribution. $\qquad\square$

### B.4 PROOF OF THEOREM 3

*Proof.* Assume the MIPS solution of $q$ is $r^*$. $r^*$ is the nearest neighbor solution of $q$ if and only if the condition below holds:

$$\|r^* - q\| < \|r - q\|, \quad \forall r \in \mathcal{D}, r \neq r^*$$

By simplifying and rearranging, we rewrite the condition as:

$$r^{*2} - r^2 < 2(\langle r^*, q \rangle - \langle r, q \rangle), \quad \forall r \in \mathcal{D}, r \neq r^*$$

Obviously, this condition does not always hold for any $q$, but we can introduce a scalar $\lambda$ to enlarge the right side of the inequality to make it hold, the solution space of $\lambda$ can be found by:

$$r^{*2} - r^2 < 2\lambda(\langle r^*, q \rangle - \langle r, q \rangle), \quad \forall r \in \mathcal{D}, r \neq r^*$$

$$\lambda > \sup\left(\left\{\frac{r^{*2} - r^2}{2(\langle r^*, q \rangle - \langle r, q \rangle)}\right\}\right), \quad \forall r \in \mathcal{D}, r \neq r^*,$$

Where $\langle r^*, q \rangle > \langle r^*, q \rangle, \forall r \in \mathcal{D}, r \neq r^*$ always holds because $r^*$ is the MIPS solution of $q$, allowing us to calculate the lower bound of $\lambda$ as above. By rearranging, this new inequality actually ensures that:

$$\|r^* - \lambda q\| < \|r - \lambda q\|, \quad \forall r \in \mathcal{D}, r \neq r^*$$

This result means search for the nearest neighbor of $\lambda q$ is equivalent to the search for the maximum inner product solution of $q$ when a proper $\lambda$ is chosen. Since the dataset $\mathcal{D}$ is finite, the lower bound of $\lambda$ can always be found.

Here, we equate the search target for the nearest neighbor of $\lambda q$ with the MIPS solution of $q$. Similarly, we can force each step of the nearest neighbor search towards $\lambda q$ being aligned with the MIPS for $q$, just by solving each local problem with above procedure and work out a new lower bound of $\lambda$'s solution space. In other words, by selecting any $\lambda$ greater than this new lower bound, we can ensure these two types of search choose the same neighbor at each step and arrive at the same solution $r^*$, such that this neighbor both minimise the Euclidean distance to $\lambda q$ and maximise the inner product with $q$. Thus, we also align the search paths of both metric under the greedy mechanism of Algorithm 1.

Given that MRNG ensures a greedy-achievable search path with Algorithm 1 from any start node $p$ to $r^*$ under the Euclidean metric (Fu et al., 2019), the $r^*$ is also greedy-achievable under the IP metric due to above alignment.

Notably, this proof also show that we do not need to work out the real bound for $\lambda$ in practice when IP metric is used to execute the MIPS. In other words, we can also use Euclidean metric to search for the MIPS solution of $q$ by searchng for the nearest neighbor solution of $\lambda q$ given a large enough $\lambda$ is set. $\qquad\square$

### B.5 PROOF OF THEOREM 4

*Proof.* By Definition 5, $\mathcal{G}^*$ includes an MRNG upon self-dominators. Moreover, each non-self-dominator node is connected to at least one self-dominator, requiring at most $o(1)$ steps for any node to reach the self-dominator node.

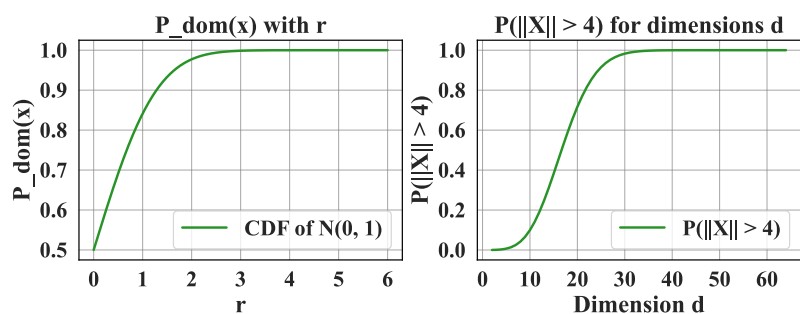

Figure 3: An illustration of how $P_{dom}(x)$ changes with $x$'s norm $r$.

Once stepping on a self-dominator node, the greedy search behaves like a search on an MRNG defined solely on self-dominators. According to (Fu et al., 2019), the expected path length between nodes in an MRNG is $\frac{cm^{1/d}\log m}{d\Delta(m)}$ with $m$ nodes in the graph, where $c$ absorbs constants. In the MRDG case, $m = \rho n$ is the number of self-dominators, where $\rho$ is the density of self-dominators. Therefore, the total expected search path length is:

$$E[\text{Path Length}] = \frac{c(\rho n)^{1/d}\log(\rho n)}{d\Delta(\rho n)} + o(1) \tag{5}$$

where $\Delta(n)$ is a slowly decreasing function of $n$ and empirically remains nearly constant as $n$ grows (Fu et al., 2019). □

## C   THEORETICAL VALIDATIONS AND IN-DEPTH ANALYSIS

### C.1   IP-DELAUNAY GRAPH AND GENERALIZED IP-DELAUNAY GRAPH

Regarding the definition of the IP-Delaunay graph, there is currently no standardized consensus in the academic community. The definition provided by iP-NSW (Morozov & Babenko, 2018) is tailored for the top-1 MIPS problem and proves that greedy search in any graph containing an IP-Delaunay graph (as per their definition) as a subgraph always converges to the exact solution of the MIPS problem. However, unlike the Delaunay graph in Euclidean space, their IP-Delaunay graph is not a globally connected graph. Thus, their definition does not extend to the top-k MIPS problem, which is crucial in current retrieval scenarios. Therefore, we have defined an IP-Delaunay graph that can query the top-k MIPS by building a strong connected graph. To clarify, we have renamed it the **Generalized IP-Delaunay Graph**.

### C.2   VISUALIZATION OF RATIO OF SELF-DOMINATORS

Figure 3 illustrates the likelihood of a vector becoming a self-dominator as its norm increases. When the norm reaches 4, the probability approaches 1, indicating that a vector is almost certain to be a self-dominator when its norm exceeds 4.

Additionally, we can estimate the proportion of self-dominators by analyzing the likelihood that a vector's norm exceeds 4, as shown in Figure 3. The figure reveals that almost all vectors have a norm greater than 4 when the dimension surpasses 40 for i.i.d. Gaussian vectors. While this may seem discouraging, most real-world datasets typically reside on manifolds with much lower intrinsic dimensions. Furthermore, the

average norm of real-world data is often not as large as that of i.i.d. multivariate Gaussian vectors, such as embeddings learned with $L_2$ regularization.

### C.3 THE RATIO OF SELF-DOMINATORS OF THE DATASETS USED IN OUR EXPERIMENTS

Table 2 presents the density of Self-Dominators across the eight datasets used in our experiments. Notably, five of these datasets exhibit a self-dominator density below 30%. As shown in Figure Figure 2. ARDG demonstrates a larger improvement on these datasets, highlighting its effectiveness when the self-dominator ratio is relatively low.

Table 2: Self-dominator ratio on eight real-world datasets.

| Datasets | Netflix | YahooMusic | MNIST | UKBench | Music100 | Text2Image1M | MNIST1M | Deep10M |
|---|---|---|---|---|---|---|---|---|
| Self dominator ratio | 3.2% | 2% | 5.4% | 97.5% | 29.3% | 99% | 6.2% | 98% |

An interesting observation is that the density of self-dominators gradually decreases as the dataset size increases, eventually stabilizing for large scales. This suggests that real-world data possesses inherent structure, which becomes more pronounced as data density increases. The decline in self-dominator density can be attributed to the fact that most points begin to fill the inner structure of the data manifold, while self-dominators are often located near the edges of the minimal convex hull that encloses the dataset. This behavior aligns with the intuition that as datasets grow, more points occupy the core of the structure, leaving fewer points as self-dominators near the boundary.

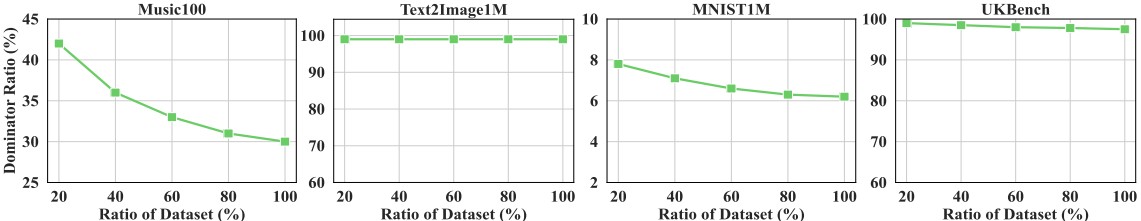

Figure 4: Self Dominator Ratio versus four million scale datasets.

## D EXPERIMENTAL DETAILS

### D.1 DATASET DETAILS

**Netflix** consists of embeddings extracted from a popular video recommender APP with Matrix Factorization (MF) (Salakhutdinov & Mnih, 2007). The dataset link is https://github.com/xinyandai/similarity-search/tree/mipsex/data/netflix.

**MNIST** is an well-known image collection of handwritten digits. We flatten the images as multi-hot representations (28 x 28 x 1), derived from https://yann.lecun.com/exdb/mnist/.

**YahooMusic** is a popular recommendation dataset uses alternating least squares for matrix factorization to obtain user and item embeddings, with item embeddings serving as dataset items and user embeddings as queries. The dataset link is https://www.cse.cuhk.edu.hk/systems/hash/gqr/dataset/yahoomusic.tar.gz

**Music100** is an audio dataset constructed from music recommender system, using IALS factorization (Hu et al., 2008) on the user-item ranking matrix to generate 100-dimensional embeddings. The dataset link is https://github.com/stanis-morozov/ip-nsw.

**UKBench** is a widely used public image search dataset for the established MIPS benchmark. The dataset link is https://www.cse.cuhk.edu.hk/systems/hash/gqr/dataset/ukbench.tar.gz.

**Text2Image1M** contains data from both textual and visual modalities, which is common for typical cross-modal retrieval tasks, where database and query vectors can potentially have different distributions in shared representation space. the database consists of image embeddings produced by the Se-ResNext-101 model, and queries are textual embeddings produced by a variant of the DSSM model. The dataset link is https://research.yandex.com/blog/benchmarks-for-billion-scale-similarity-search.

**MNIST1M** is a large-scale version of MNIST, where new images are generated through data augmentation. We use the infinite MNIST project to generate the dataset https://leon.bottou.org/projects/infimnist.

**Deep10M** consists of 10M image embeddings produced as the outputs from the last fully-connected layer of the GoogLeNet model, which was pretrained on the Imagenet classification task. The embeddings are then compressed by PCA to 96 dimensions and l2-normalized. The dataset link is https://research.yandex.com/blog/benchmarks-for-billion-scale-similarity-search

## D.2 INDEXING PERFORMANCE

The indexing time and resulting index sizes for all methods across the eight datasets are summarized in Table 3. These metrics highlight key trade-offs between indexing speed and memory efficiency. While some methods, like Fargo and ScaNN, demonstrate faster indexing times due to simpler algorithmic structures (near $O(nd)$ complexity), they compromise significantly on search performance, especially for large-scale datasets. In contrast, graph-based methods, including ARDG, exhibit longer indexing times but offer superior search accuracy and scalability. Notably, ARDG achieves a balance between speed and memory usage, outperforming most other graph-based methods in both indexing time and index size. Its sparse structure results in an index size approximately $2\times$ smaller than competitors, such as ip-NSW and ip-NSW+, making it both memory-efficient and highly effective for large datasets. These results underscore ARDG's cost-effectiveness, delivering top-tier search performance with relatively low memory and indexing overhead.

Table 3: Indexing time and sizes of different methods across eight datasets.

| Datasets | | Fargo | ScaNN | ip-NSW | ip-NSW+ | Möbius-Graph | NAPG | IPDG | ARDG |
|---|---|---|---|---|---|---|---|---|---|
| Netflix | Index Size | 2 (MB) | 8 (MB) | 7 (MB) | 6 (MB) | 5 (MB) | 7 (MB) | 7 (MB) | 3 (MB) |
| | Indexing Time | 2 (s) | 4 (s) | 12 (s) | 10 (s) | 3 (s) | 15 (s) | 94 (s) | 10 (s) |
| MNIST | Index Size | 18 (MB) | 60 (MB) | 23 (MB) | 28 (MB) | 16 (MB) | 23 (MB) | 20 (MB) | 12 (MB) |
| | Indexing Time | 3 (s) | 33 (s) | 30 (s) | 41 (s) | 32 (s) | 54 (s) | 492 | 45 (s) |
| YahooMusic | Index Size | 27 (MB) | 174 (MB) | 53 (MB) | 87 (MB) | 53 (MB) | 54 (MB) | 53 (MB) | 27 (MB) |
| | Indexing Time | 2 (s) | 23(s) | 38 (s) | 55 (s) | 17 (s) | 40 (s) | 447 (s) | 37 (s) |
| UKBench | Index Size | 38 (MB) | 1205 (MB) | 423 (MB) | 767 (MB) | 423 (MB) | 423 (MB) | 423 (MB) | 213 (MB) |
| | Indexing Time | 4 (s) | 98 (s) | 296 (s) | 600 (s) | 87 (s) | 405 (s) | 1648 (s) | 220 (s) |
| Music100 | Index Size | 40 (MB) | 566 (MB) | 386 (MB) | 466 (MB) | 363 (MB) | 401 (MB) | 566 (MB) | 195 (MB) |
| | Indexing Time | 7 (s) | 30 (s) | 258 (s) | 1075 (s) | 207 (s) | 326 (s) | 1700(s) | 107 (s) |
| Text2Image1M | Index Size | 46 (MB) | 760 (MB) | 385 (MB) | 466 (MB) | 385 (MB) | 385 (MB) | 408 (MB) | 196 (MB) |
| | Indexing Time | 11 (s) | 120 (s) | 539 (s) | 1160 (s) | 436 (s) | 550 (s) | 1969 (s) | 280 (s) |
| MNIST1M | Index Size | 52 (MB) | 1006 (MB) | 402 (MB) | 682 (MB) | 389 (MB) | 430 (MB) | 389 (MB) | 204(MB) |
| | Indexing Time | 12 (s) | 360 (s) | 1860 (s) | 3320 (s) | 302 (s) | 2544 (s) | 6682 (s) | 459 (s) |
| Deep10M | Index Size | 343 (MB) | 7428 (MB) | 3850 (MB) | 4659 (MB) | 3850 (MB) | 3850 (MB) | 3850 (MB) | 1946 (MB) |
| | Indexing Time | 47 (s) | 448 (s) | 4383 (s) | 9602 (s) | 4658 (s) | 5872 (s) | 16830 (s) | 3789 (s) |

Table 4: The indexing performance over various data scales on Deep10M.

| Dataset Scale | Indexing Time | Index Size |
|---|---|---|
| Deep100K | 29 (s) | 20 (MB) |
| Deep1M | 340 (s) | 195 (MB) |
| Deep10M | 3789 (s) | 1946 (MB) |

## D.3 SCALABILITY EXPERIMENTS

Scalability in MIPS methods encompasses two key aspects: scalability with the dataset size $n$ and scalability with the required number of solutions $k$. Figure 5 demonstrates ARDG's scalability with $n$, showing a near $O(\log n)$ growth rate for search performance, which is highly efficient for large-scale datasets. Table 4 further highlights the indexing scalability, with ARDG maintaining a near $O(nd \log n)$ growth rate during the indexing process, aligned with our analysis in Section 4.2. Impressively, ARDG can index a dataset as large as Deep10M (10 million data points) in approximately one hour, showcasing its practical efficiency in handling large datasets. Additionally, Figure 7 illustrates ARDG's scalability with respect to the required solution quantity $k$, revealing a near-linear $O(k)$ growth rate for $recall@k$. This growth rate is manageable due to the relatively small overhead, with the recall@100 on Deep10M requiring only 0.4 ms per query. These results emphasize ARDG's ability to efficiently scale both in terms of dataset size and the number of required solutions, making it a robust choice for large-scale applications.

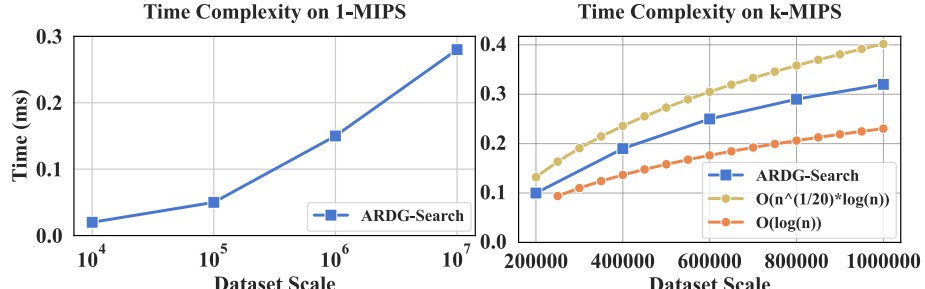

Figure 5: Query time versus data scale on Deep10M dataset at 98% recall for top-1 and top-$k$ MIPS.

## D.4 WORST CASE ANALYSIS IN TIME COMPLEXITY

Consider a scenario in very high-dimensional spaces where all data points lie on a great circle (the intersection of a hyperplane passing through the origin and a hypersphere centered at the origin). In this case, each point on this curve becomes a self-dominator, dominating only itself. When sparsifying the graph, both MRDG and ARDG will, with high probability, link each node to only its two adjacent neighbors on the curve. Consequently ARDG will fail to accelerate search on such data, resulting in a linear search complexity of $O(nd)$. Additionally, constructing the ARDG incurs an approximate indexing complexity of $O(n^2 d \log n)$, since ARDG relies on other MIPS methods to construct the base $k$-MIP graph. In this scenario, all other MIPS methods also fail to accelerate MIPS on this data. However, such extreme cases hardly occur in practice.

### D.5 PARAMETER SENSITIVITY

The left of Figure 6 illustrates how the search performance for both top-1 MIPS and top-100 MIPS varies with the balancing factor $\alpha$. This parameter effectively balances the trade-off between locality and global connectivity, ensuring that both types of searches are optimized. As $\alpha$ decreases, the algorithm improves transitions to self-dominators (benefiting top-1 MIPS), while lower values of $\alpha$ enhance local neighborhood connectivity, which is crucial for top-100 MIPS.

The right of Figure 6 highlights that increasing the parameter $k$ for the initial $k$-MIP graph improves search performance. However, this improvement quickly plateaus, indicating that only a moderately large $k$ ($k = 200$) is needed to capture sufficient local structure for efficient search. This suggests that excessively large values of $k$ offer diminishing returns, allowing for an optimal balance between search performance and indexing cost.

Finally, Figure 8 demonstrates that the maximum out-degree $R$ plays a pivotal role in search efficiency. The results reveal a concave curve, indicating the existence of an optimal value for $R$. Fortunately, finding this optimal value is straightforward, as search performance is relatively stable and smooth near the optimal point. This further underscores ARDG's ability to achieve efficient search with minimal tuning.

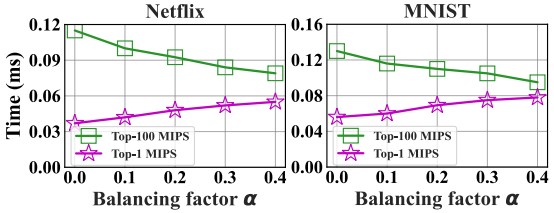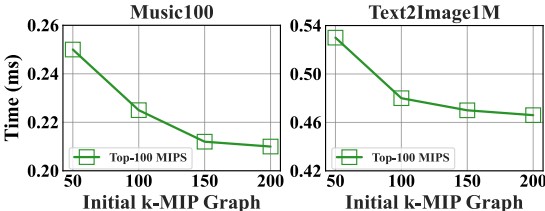

Figure 6: Query time versus different balancing factor $\alpha$ (Left) and $k$-MIP graph (Right).

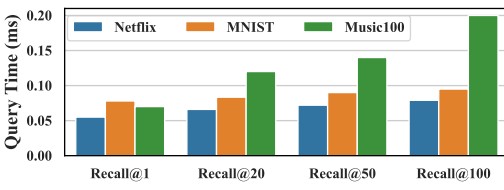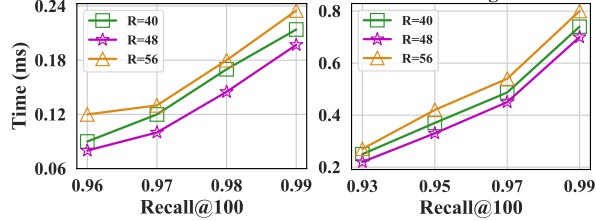

Figure 7: The query time of various k of recall on Netflix, MNIST and Music100 at 99% recall.

Figure 8: The query time versus different out degree R.

### D.6 EXPERIMENTS ON COUNTING DISTANCE COMPUTATIONS

While the primary experiments use query execution time as the evaluation metric to ensure fair comparisons with hashing and quantization-based methods, the number of Distance Computations (DC) is the most informative metric for evaluating graph-based methods. This is because DC predominantly determines the search efficiency of graph-based methods and shields interference of system factors, particularly in high-dimensional spaces. A lower number of DCs implies faster graph traversal and reduced computational overhead per step. As shown in Figure 9, ARDG significantly outperforms other methods in terms of DCs, further validating the efficiency of its design and the soundness of our theoretical findings. This reduc-

tion in DC confirms that ARDG's sparse yet well-connected structure allows for efficient navigation while minimizing redundant computations.

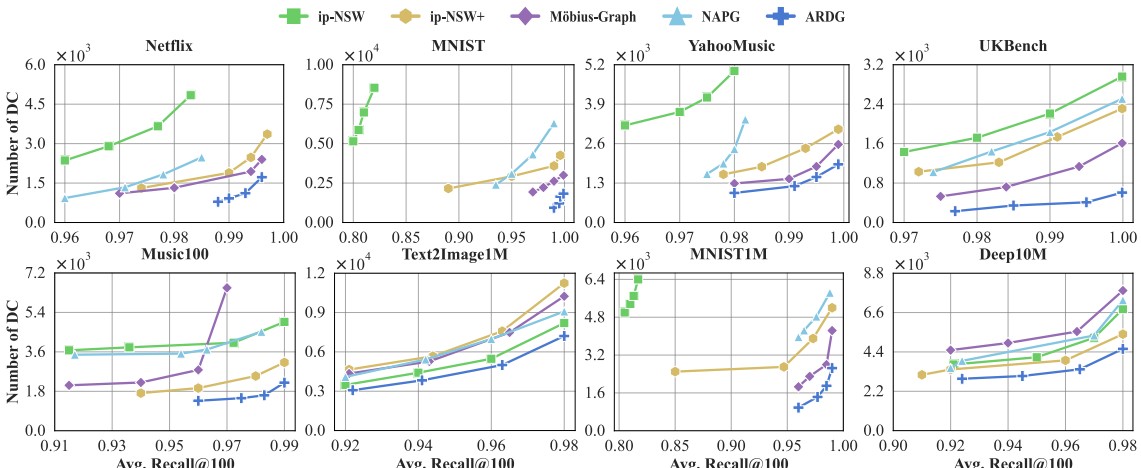

Figure 9: Experimental results of Distance Computation on eight datasets. Lower right is better.

## D.7 SEPARATE VIEW OF METHODS ABSENT FROM MAIN EXPERIMENTAL RESULTS

Figure 10 presents the performance of methods that lag behind the top-tier approaches in our main experimental results. These methods, including Fargo and ScaNN, often exhibit performance bottlenecks due to limitations in their indexing or search algorithms. While they may perform adequately in smaller datasets or lower dimensions, they struggle with scalability in high-dimensional and large-scale datasets. The separate view highlights their inability to compete with graph-based methods like ARDG, especially in terms of precision and recall under higher-dimensional conditions.

Figure 11 illustrates the query performance of IPDG, which suffers due to its graph's poor connectivity, resulting in weaker performance across various datasets. This highlights the superior scalability and efficiency of ARDG compared to other graph methods.

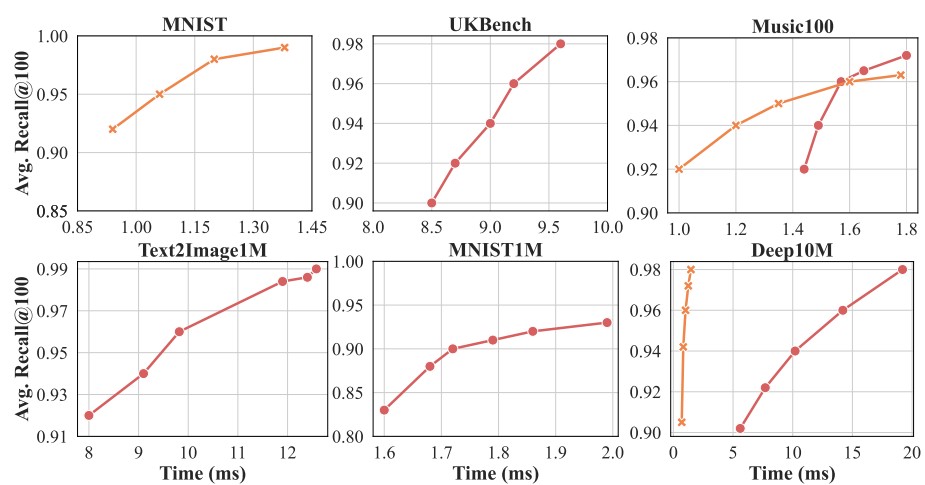

Figure 10: Separate View of Fargo and ScaNN from Main Results. Upper left is better.

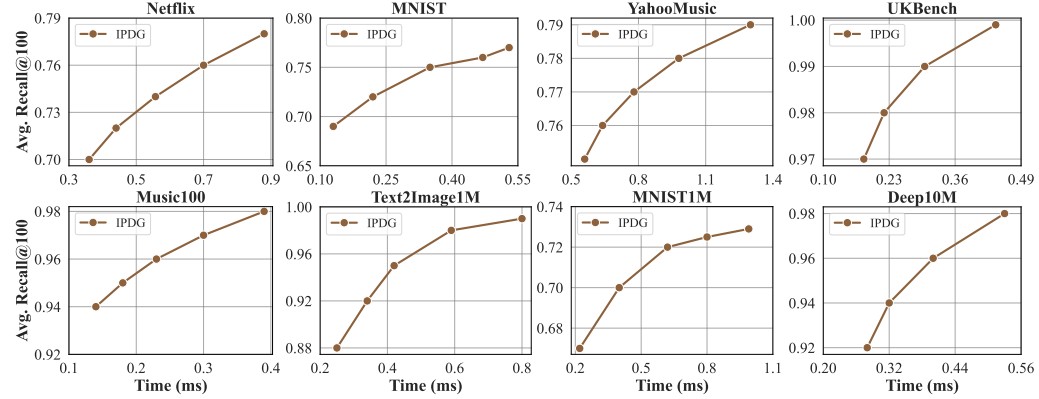

Figure 11: Separate View of IPDG from Main Results.

