# OpenReview forum: "Identify Dominators: The Key To Improve Large-Scale Maximum Inner Product Search"
_ICLR.cc/2025/Conference — Submitted to ICLR 2025_

### Official Review · Reviewer_6cFh · 2024-10-16

**Soundness:** 4
**Presentation:** 4
**Contribution:** 3
**Rating:** 6
**Confidence:** 4

**Summary:**

This paper studies a graph-based algorithm for maximal inner product search (MIPS). Compared to previous solutions that reduce MIPS to nearest neighbor search (NNS), this paper leverages the intrinsic geometry of the inner product space. The authors provide a theoretical running time analysis for randomly generated data. Experiments verify that the proposed method provides better query performance score (QPS) compared to other baselines by a significant margin.

**Strengths:**

1. I appreciate the authors for choosing a meaningful yet often overlooked problem, as maximal inner product search (MIPS) is usually reduced to nearest neighbor search (NNS).
2. Both theoretical and empirical results are provided for the proposed method. I especially appreciate the worst-case analysis in the appendix, as many similar papers on graph-based search algorithms often overlook their worst-case behaviors. You can find some similar works that focus on the worst-case analysis of NNS. Regarding the experimental results, I commend the authors for their efforts in including a diverse range of datasets and baselines.
3. The paper is clearly written and easy to follow.

**Weaknesses:**

1. The algorithmic framework is not novel compared to many graph-based algorithms for nearest neighbor search (NNS). For example, it first proposes a new graph with a defined property, proves some convergence bounds under random data distribution, and then adds heuristics to make it empirically fast.

**Questions:**

n/a

---

> ### Author Response · Authors · 2024-11-19
>
> We appreciate the reviewer' agreement with the significance of the problem, as well as the theoretical and experimental results. Below, I will elaborate on the clarification of the novelty.
>
> (1) The development of graphs in the MIPS field currently faces limitations. Their methods either reduce the problem to NNS but often are constrained by specific data distributions [1,2], or they directly perform heuristic edge pruning in the IP space, lacking theoretical validation [3,4].
>
> (2)We are committed to proposing a theoretical graph within the IP space and designing an edge-pruning scheme based on its theoretical properties. Our theory seamlessly integrates with our pruning algorithm, providing a reliable and practical pruning strategy rather than a purely heuristic one. The pruning method is theoretically supported, and easy to execute. The search complexity can approach the theoretical complexity, thereby achieving good performance on various MIPS retrieval datasets.
>
> (3) Our algorithm can provide an efficient and robust retrieval index for various downstream tasks, making it easier for real users to utilize. In the field of retrieval, query performance advantages are paramount, and our improvements hold significant performance value. This has also been widely recognized by the reviewers.
>
> [1] FARGO: Fast Maximum Inner Product Search via Global Multi-Probing. VLDB, 2023.
>
> [2] Möbius Transformation for Fast Inner Product Search on Graph. NeurIPS, 2019.
>
> [3]Non-metric Similarity Graphs for Maximum Inner Product Search. NeurIPS. NeurIPS, 2018.
>
> [4]Norm Adjusted Proximity Graph for Fast Inner Product Retrieval. KDD, 2021.

---

### Official Review · Reviewer_nF12 · 2024-11-01

**Soundness:** 1
**Presentation:** 2
**Contribution:** 3
**Rating:** 1
**Confidence:** 5

**Summary:**

The article proposes a graph method called Approximate Relative Dominator Graph (ARDG) for approximate maximum inner product search (MIPS). The article begins by exploring the properties of the Voronoi diagram and its dual, Delaunay graph, when defined under the inner product (IP) instead of the Euclidean distance. The authors call a database point that belongs to its own IP-Voronoi region (the IP-Voronoi region corresponding to the database point $x_i$ is a region of points whose inner product with $x_i$ is larger than their inner product with any other database point) a self-dominator. The proposed method is based on (approximately) identifying the self-dominators, and building a graph that maintains sparse connections between them. According to the empirical results, the proposed method outperforms the earlier MIPS methods.

**Strengths:**

The exploration of the geometric properties of the inner product is interesting. The indexing methodology is novel as far as I know. According to the experiments, the proposed method performs well.

**Weaknesses:**

There are so many unclear parts and outright errors in the article that I find it very hard to read. For instance:

- In the illustration of IP-Delaunay graph in Figure 1 (b), there are connections between some of the ordinary (non-dominating) points that belong to the adjacent IP-Voronoi cells. But these connections do not follow from Definition 2 (according to Definition 2, there should only be connections between the dominators of the adjacent cells, and between a point and the dominator of the cell that point belongs to).
-  The definition of the IP-Delaunay graph (Definition 2) is different than the definition given in the earlier literature, but no explanation is given for this different definition (see Questions).
- In Figures 1 (c) and Figures 1 (d) the point at approximately (-0.4,-0,8) is an out-dominator and should be dark green.
- In Figure 1 (g) labels refer to out-dominators 4 and 5 that do not exist.
- The claim of Theorem 1 is not well-defined (see Questions).
- Figure 1 (c) claims to illustrate the NDG graph built by the algorithm described in the claim of Theorem 1. However, for any point $x_i$, the algorithm given in Theorem should produce edges between $x_i$ and all the self-dominators $\mathcal{S}_{dom}$. In Figure 1 (c), there is only an edge between $x_i$ and the dominator of the Voronoi cell the point $x_i$ belongs to (In addition, there is an edge between the point at (-0.4,-0,8) and the out-dominator. This edge should not exist according to the algorithm given in Theorem 1).
- Given Definition 1, Definition 3 is superfluous. You should remove Definition 3 and simply say in Definition 1 that the associated vector $x$ is called a dominator.
- The claim of Theorem 2 does not hold. Clearly the dataset size $n$ affects the probability that a dataset point with a given norm is a self-dominator, and this probability goes to zero as $n$ grows to infinity by the law of large numbers. You are computing a probability that _one_ random vector whose components are drawn i.i.d. from $N(0,1)$ dominates a point with a norm $r$.

These examples are only the ones I found when carefully reading the first couple of pages of material. Based on this, I can only assume that the rest of the article is of equally low quality.

In addition to the errors in the theoretical material, the empirical evaluation is insufficient. The authors admit that the algorithms are not fine-tuned for the benchmark data sets and claim that this favors algorithms with low parameter sensitivity. However, I do not think this is a fair setting for a comparison. This is because naturally authors (in general) are always more aware of the sensible hyperparameter settings of their own algorihm (and spend more time and effort for choosing the hyperparameters) compared to the baselines. Consequently, baseline methods are often tested at sub-optimal hyperparameter combinations, which distorts the results. For instance, in this case the authors state that the baselines Fargo and ScaNN are absent from some figures (e.g., Deep10M) due to instability that worsens as cardinality and dimensionality increase. But this is almost surely because of insufficient hyperparameter grids. The benchmark data sets are small or moderate-sized, and ScaNN should have no issue at all with these data sets if hyperparameters were even in the ballpark.

Thus, I see that there are only two ways to ensure a fair performance comparison: (1) to enable the authors of the baseline algorithms to select their own hyperparameter grids, as is done in ANN-benchmarks (Aumüller et al., 2020) that is the gold standard for the performance evaluation in the field; or (2) to carefully tune the hyperparameters of the baseline methods to ensure that the algorithms are compared at near-optimal hyperparameter settings.

In summary, the technical quality of the article is very low, and it is clearly sent unfinished. There is no way the article can be accepted for this conference. However, the approach is interesting and the results seem promising, so I recommend that authors continue working on the manuscript.

Aumüller, Martin, Erik Bernhardsson, and Alexander Faithfull. "ANN-Benchmarks: A benchmarking tool for approximate nearest neighbor algorithms." Information Systems 87 (2020): 101374

**Questions:**

(1) What you exactly want to prove in Theorem 1? I cannot parse the claim. The theorem claims that Naive Dominator Graph (NDG) can be build by a given algorithm, but NDG is not defined anywhere. Or is the claim that the graph built by the algorithm (and that you call NDG) given in the claim is strongly connected? If this is the case, what means a strongly connected NDG $\mathcal{G}$ on the dominator set $\mathcal{P}$? I know what is a connected graph, but i do not know what is the meaning of a connected graph on a subset of nodes. Do you mean that the nodes $\mathcal{S}_{\mathrm{dom}}$ are connected to each other? Finally, strong connectivity is a property of directed graphs, whereas you build an undirected graph (you should simply say that the graph is connected).

(2) Why do you redefine IP-Delaunay graph (Definition 2) differently from earlier literature (Morozov & Babenko 2018)? You add an edge between a point $x_i$ and the dominator of the Voronoi cell the point $x_i$ belongs to. Morozov & Babenko (2018) prove (Corollary 1) that greedy search in any graph that has an IP-Delaunay graph (their definition) as a subgraph always converges to the exact solution of the MIPS problem. Hence, what is the motivation of adding extra edges?

Morozov, Stanislav, and Artem Babenko. "Non-metric similarity graphs for maximum inner product search." Advances in Neural Information Processing Systems 31 (2018).

---

> ### Author Response · Authors · 2024-11-19
>
> We thank the reviewer for their valuable feedback. Below, we provide clarifications.
>
> **W1. The problem with Figure 1.**
>
> We appreciate the reviewer's suggestions and have corrected the typos in Figure 1 to more clearly illustrate our motivation.
>
> (1) Figure 1(b) illustrates an example of the IP-NSW graph, from which it is evident that IP-NSW contains some redundant edges.
>
> (2) We have corrected the coordinates of one node in the figure to ensure it has the correct label.
>
> (3) We have fixed the legend in Figure 1(g).
>
> (4) We have defined the NDG and refined Figure 1(d) accordingly.
>
> **W2. The claims of certain Theorems and Definitions are not well-defined.**
>
> (1) For Definitions 1 and 3, we aim to use Definition 3 to define the concept of a dominator, which will then lead to the theoretical analysis in the subsequent sections. This approach enhances the theoretical coherence of our work. **We have improved the Definition 3 to avoid redundancy.**
>
> (2) Regarding the definition of the IP-Delaunay graph, there is currently no standardized consensus in the academic community. The definition provided by IP-NSW[1] is tailored for the top-1 MIPS problem and proves that greedy search in any graph containing an IP-Delaunay graph (as per their definition) as a subgraph always converges to the exact solution of the MIPS problem. We acknowledge this point. However, unlike the Delaunay graph in Euclidean space, their IP-Delaunay graph is not a globally connected graph. **Their definition does not extend to the top-k MIPS problem, which is crucial in current retrieval scenarios.** Therefore, we have defined a Generalized IP-Delaunay graph that can query the top-k MIPS. To clarify, we have renamed it the Generalized IP-Delaunay Graph.
>
> (3) For Theorem 1, to more clearly illustrate the NDG, we have decomposed it into a definition (New Definition 4) and a theorem (New Theorem 1). First, we define the NDG based on the edge pruning rules. Subsequently, we use Theorem 1 to prove three properties of the NDG: (1) it is a strongly connected directed graph, (2) it can identify all dominators in the dataset, and (3) it includes a fully connected graph among all dominators. These properties lead us to introduce two types of dominators and deduce that the NDG can accurately obtain the top-k MIPS. These improvements will allow the reviewers to more clearly understand the content and role of the theorem.
>
> [1] Non-metric similarity graphs for maximum inner product search, NeurIPS, 2018.

---

> ### Author Response · Authors · 2024-11-19
>
> **W3. The concern about fair comparison.**
>
> Our decision to use uniform parameters was driven by practical considerations, acknowledging the diverse modalities and scales present in real-world data. Users often lack the experience to select optimal parameters for retrieval tasks, making the low-sensitivity algorithm beneficial. It ensures consistent and ideal retrieval results across various scenarios, thereby enhancing user experience.
>
> Furthermore, we have also fine-tuned the relevant parameters offline (to see the difference between various parameters). For graph algorithms, the most critical universal parameters are EFC (edge selection candidate count) and R (node degree), as they directly influence the index's quality. Empirically, a larger EFC generally improves performance by expanding the candidate node set, which aligns more closely with theoretical edge pruning. Conversely, R behaves as a convex function, with an optimal value that varies across datasets.
>
> To validate our approach, we evaluated two representative graph baselines (ip-nsw and mobius graph) under different parameters (EFC-R) and datasets (ukbench, deep10m). The results showed that query speed variations with differing construction parameters were minimal (up to 6%), confirming the algorithms' low sensitivity and the robustness of our experimental setup. This low sensitivity ensures that our findings remain valid and applicable across a wide range of scenarios.
>
> |                          |            |            |            |            |
> | ------------------------ | ---------- | ---------- | ---------- | ---------- |
> | UKBench(recall@100=0.99) | **256-24** | **256-32** | **512-32** | **512-48** |
> | ip-NSW                   | 0.36       | 0.345      | 0.329      | 0.337      |
> | Mobius Graph             | 0.253      | 0.246      | 0.244      | 0.240      |
>
> | Deep10m (recall@100=0.98) | **256-24** | **256-32** | **512-32** | **512-48** |
> | ------------------------- | ---------- | ---------- | ---------- | ---------- |
> | ip-NSW                    | 0.561      | 0.548      | 0.543      | 0.529      |
> | Mobius Graph              | 0.737      | 0.728      | 0.722      | 0.695      |
>
> 2. SCANN is sensitive to data scale and dimensionality. On the million-scale dataset, we used the official recommended parameters [1], while on the deep10M dataset, we conducted further fine-tuning to scale the setting. The quantization algorithm did not encounter a performance bottleneck on deep10M; it achieved 98% accuracy in 1.2ms (Appendix.D7). However, there is a noticeable gap (2x) compared to sota graph algorithms, so we moved it to the appendix. While we cannot guarantee these are the optimal parameters, we believe they are relatively fair. As shown in the relevant papers [2,3,4], there is a speed gap between quantization algorithms and graph-based algorithms, although the former has its advantages. For any unclear expressions, we have revised them in the text. We also look forward to receiving feedback from the reviewer.
>
> [1] https://github.com/google-research/google-research/tree/master/scann
>
> [2] Reconsidering Tree based Methods for k-Maximum Inner-Product Search: The LRUS-CoverTree. ICDE, 2024.
>
> [3] Survey of vector database management systems. VLDBJ, 2024
>
> [4] A Comprehensive Survey and Experimental Comparison of Graph-Based Approximate Nearest Neighbor Search. VLDB, 2021.

---

> > ### Comment · Reviewer_nF12 · 2024-11-25
> >
> > Thank you for the detailed response.
> >
> > I appreciate the efforts for improving the writing and quality of the article. Unfortunately my main objection was the unacceptably low technical quality (the other reviewers noticed even more mistakes in addition to the ones I pointed out in my review) of the original submission, so I will maintain my score.  In addition, you have not addressed Theorem 2, whose claim seems to be obviously wrong, in your rebuttal.

---

### Official Review · Reviewer_JPSr · 2024-11-02

**Soundness:** 3
**Presentation:** 3
**Contribution:** 3
**Rating:** 6
**Confidence:** 5

**Summary:**

This paper proposes a graph-based ANN method for MIPS. The experiments on public datasets show that the proposed method achieves a 30% average speedup in search at high precision compared to state-of-the-art graph-based methods.

**Strengths:**

S1. This paper proposes a simple but effective method for selecting edges in proximity graph. The procedure adapts the edge selection procedure in HNSW to the MIPS context.
S2. The discussion of self-dominator makes the proposed method intuitive.
S3. The empirical evaluations show that the proposed method outperforms baselines.

**Weaknesses:**

W1. Line 130: "the Mobius transformation normalizes each vector p \in R^d to p/|p|". This is incorrect. In Zhou et al., 2019, it should be p/(|p|^2). The Mobius transformation is not a normalization, instead, it maps the vectors with larger norms closer to the zero point for faster searching in the proximity graph.

W2. The author claims the transformation method introduces data distortion, which is not clearly explained. I would suggest the author to illustrate why this distortion is problematic. For example, followed by W1, the Mobius transformation is only applied on the data during the indexing stage. On the searching time, original data is employed to compute the inner product. What are the disadvantages of this transformation?

W3. Algorithm 1 is unclear and incorrect. R should be a min heap instead of max heap because it maintains the current most similar vector ids and always pops out the worst one within the heap. The batch_insert function is not explained.

W4. The experiments is slightly flawed. "For query execution, we disable additional compiler optimizations and use same number of threads to ensure a fair comparison" What is the additional compiler optimizations? Since the query time is the main comparison in the experiment, I assume we should enable all optimizations (which reduces the implementation gaps) to show

**Questions:**

See weakness.

---

> ### Author Response · Authors · 2024-11-19
>
> We thank the reviewer for their valuable feedback. Below, we provide clarifications.
>
> **W1. The author claims the transformation method introduces data distortion, which is not clearly explained. I would suggest the author illustrate why this distortion is problematic.**
>
> The isomorphism derived from the Möbius transformation is predicated on strong assumptions [1], including the requirement that the origin point lies within the dataset's convex hull and that the data is independently and identically distributed (IID). These assumptions are frequently violated in real-world scenarios. The IID assumption, in particular, is often breached due to the inherent correlations among the representations generated by machine learning models across different dimensions [2,3]. When these assumptions are not satisfied, the vector transformation can falter, as it fails to approximate the true IP-Delaunay graph using Euclidean distances.
>
> **W2. Algorithm 2 is unclear and incorrect.**
>
> Thank you for the feedback. We have revised the content of Algorithm 1 to enhance clarity. Specifically, we have adjusted R to function as a min-heap and provided a detailed explanation of the batch search process.
>
> **W3. What are the additional compiler optimizations?**
>
> Query time is our foremost concern, and we strive to ensure that the algorithm's query performance is as unaffected as possible by extraneous factors.
>
> By "additional compiler optimizations," we mean optimizations performed by the compiler that alter the code itself rather than the algorithm, such as "Loop Unrolling." Given that vector computation is the bottleneck in the query process, we employed the same vector computation function and disabled components specifically designed to accelerate vector computation. We believe this approach reduces the implementation gap.
>
> [1] Möbius Transformation for Fast Inner Product Search on Graph, NeurIPS, 2019.
>
> [2] Learning transferable visual models from natural language supervision, ICML, 2021.
>
> [3] An open large-scale dataset for training next generation image-text models. NeurIPS, 2022.

---

> > ### Comment · Reviewer_JPSr · 2024-11-26
> >
> > I appreciate the author's clarifications. I would like to keep my score unchanged.

---

### Official Review · Reviewer_osTr · 2024-11-02

**Soundness:** 2
**Presentation:** 2
**Contribution:** 2
**Rating:** 5
**Confidence:** 4

**Summary:**

The paper is on the Maximum Inner Product Search (MIPS) problem in high-dimensional  spaces.  This is a well-motivated "nearest-neighbor"-type (NN) problem in IR and related fields.  A common way to solve it is to reduce it to a standard NN problem on the Euclidean space, which is far better understood.  But this is not good in many settings.  The paper aims to study the problem directly in the Hilbert space.  Note inner product is a similarity metric and not a distance metric.  Hence properties such as the triangle inequality do not hold in this inner product space.  The main tool in the paper is to explore the Voronoi cells and the associated Delauney graph, induced by the data.

The paper has two contributions from a technical point of view.  The first is to identify self-dominators as an important subset of data for MIPS purposes; there is natural associated graph called Monotonic Relative Dominator Graph (MRDG).  The second is to approximate MRDG by edge pruning to obtain ARDG - which has better computational properties.  It reduces the indexing costs from $O(d n^2)$ to
 $O(nd (\log n + r^2))$, where $r$ is max of 2-hop neighborhood size.

The paper performs detailed experiments on public data to show 30% speed up over existing methods.

**Strengths:**

* The MIPS problem is an important problem in information retrieval and NLP settings
* The paper shows good experimental results - a 30% improvement on public datasets over some existing methods is a decent practical win

**Weaknesses:**

* The novelty is on the low side.  The paper is not the first one to study the geometry of the IP space.  For example, the paper of Mozorov and Babenko (NIPS 2018) studied the Voronoi cells and their properties.  Algorithm 1 is from there.  It it likely that there are even earlier references.  The paper also uses observations from the prior work of Liu et al (AAAI 2020) and Tan et al (EMNLP 2019), to motivate self-dominators.  The MRNG notion seems to be from Fu et al. (VLDB 2019).

* The theoretical contributions are not compelling.  Theorem 2 is a straightforward and special case.  Theorems 3 & 4 are equally straightforward, with Theorem 4 largely built upon Fu et al.

* The paper is based on heuristic approaches for a problem where nice algorithmic ideas are likely to exist (as seen in some previous works).  The theoretical contributions can be completely discounted and the paper's merits stands only on the practical wins.

* The paper suffers from extremely poor writing.  There are many typos and overstated claims in the text.

Theorem 2: overclaimed - special case - vectors are iid and normal.

Main contribution:

In what way ARDG approximates MRNG?

Minor comments:

l22: theoretical solid foundations -> solid theoretical foundations

l39: overstatement: deep understanding of the geometric properties of the inner product (IP) space remains elusive

l53: strongly connected to self-dominators?  please explain

l56: what is the expectation over?

l57: what is $n$?

l74: preliminary -> preliminaries

l77: represents -> represent

l89: "IP is not a typical metric": what is typical

l135: frequent index rebuilding: unclear what is meant here

l145: $x_i$ is contained in $V_{x_j}$; Need to explain why this asymmetry while the edge you add is bidirectional.  Also explain if $x_i$ contained in $V_{x_j}$ woudl imply $V_{x_i} \subseteq V_{x_j}$?  IP Voronoi cells behave differently from Euclidean Voronoi cells and it is better to explain things clearer since most readers will be familiar with the latter.  For eg, in the case of inner product the Voronoi cells for some points are empty.

l173: exloring -> exploring

l175: Something that could be clarified with this definition: if $V_x$ is the Voronoi cell of $x$, then by Definition 1, isn't it the dominator as well?  Ie, the conditions in Definition 1 and Definition 3 look identical.

l177: "finite full coverage of": explain what you mean

l178: ${\cal S}_{dom}$ dominator set never defined

l183: Why is this a Theorem. This seems like a definition.

l191: should it be $||x||^2$?

l219: overclaimed: this statement assumes that the dataset comes from the iid normal.  The text is written as if it holds for all datasets.

l328: How do you justify the assumption that $r$ is a constant?

l461: overclaimed: Many other papers have investigated this problem

l703: Proof - doesn't these follow from prior work?

**Questions:**

What is the notion of approximation are you using for ARDG?  It appears to be just a heuristic.

---

> ### Author Response · Authors · 2024-11-19
>
> We thank the reviewer for their valuable feedback. Below, we provide clarifications.
>
> **W1. The novelty is on the low side. The theoretical contributions are not compelling.**
>
> In this paper, we aim to develop an efficient and practical retrieval algorithm tailored for downstream tasks. While we acknowledge that we are not the first to explore geometric aspects within the IP space, we contend that the unique properties of the inner product space, distinct from the Euclidean space, render existing methods theoretically unsupported. This lack of theoretical grounding results in suboptimal search performance.
>
> **We clarify the choice of the Gaussian distribution for theoretical analysis**:
>
> Given the diverse distributions of high-dimensional spaces across various datasets, it is challenging to conduct theoretical analysis on general data. In the realm of theoretical exploration within the vector search field, it is both reasonable and widely accepted to make prior assumptions about data distribution [1,2,3,4]. Among these assumptions, the Gaussian distribution is a prevalent choice, as it more closely mirrors the data distribution observed in the real world.
>
> **We clarify and highlight our contributions as follows**:
>
> 1. Our theory seeks to illustrate the efficiency of graph structures in IP retrieval. Theorem 3 establishes reachability, while Theorem 4 demonstrates search complexity; these two theorems are crucial and offer theoretical backing for IP retrieval. We could leverage the properties of MRNG to derive complexity because we **constructed a bridge between the IP and NN spaces**, which is a non-trivial achievement. Previous work has yet to explore this aspect within the IP space.
> 2. We introduce a novel edge pruning rule that harnesses the geometric properties of the IP space to build a compact index. This is a non-trivial innovation. Analogous to how HNSW[5] and NSG [1] algorithms employ the MRNG edge pruning rule for index construction, this rule also forms a pivotal motivation for their algorithmic contributions.
> 3.  Our theory seamlessly integrates with our pruning algorithm, providing a reliable and practical pruning strategy rather than a purely heuristic one. In contrast to other heuristic algorithms in the IP space, our edge pruning method is both efficient and aligns with the theoretical graph, resulting in enhanced search performance. We have experimentally validated its efficiency in construction and its conformity with search complexity (Appendix.D3).
> 4.  In the field of retrieval, query performance advantages are paramount, and our improvements hold significant performance value. This has also been widely recognized by the reviewers.
>
> **W2. The typos, overstated claims, and minor comments should be fixed.**
>
> Thanks for pointing it out, and we fixed the writing typos. We agree there are some overstated claims, we have revised the relevant sections to make them more rigorous. Here we list some important revisions:
>
> 1. We have revised and reclaimed the relationship between this paper and related work.
> 2. We have expanded on the relevant terms to enhance readability and ensure clarity.
> 3. We have revised certain items that might cause confusion in the revised paper (e.g., IP-Delaunay Graph)
>
> **Q1. What is the notion of approximation are you using for ARDG?**
>
> (1) In the realm of graph index literature, "approximation" denotes our method of creating an approximate graph that **preserves as much information as possible from the theoretical graph, thereby achieving strong search performance with reduced index complexity** [1,2,3,4]. This term does not refer to the lower bound of a specific algorithm.
>
>  (2)In our approach, we initially constructed a k-MIP graph to generate high-quality candidates. Then, guided by theoretical principles, we employ an edge-pruning strategy to ensure the approximate graph retains the geometric properties of the theoretical graph while maintaining sparsity. Our practical complexity experiments, detailed in Appendix D3, show that **our approximate graph aligns well with the theoretical graph, validating its approximation quality.**
>
> [1] Fast Approximate Nearest Neighbor Search With The Navigating Spreading-out Graph, VLDB,2019.
>
> [2] Efficient Approximate Nearest Neighbor Search in Multi-dimensional Databases, SIGMOD, 2023.
>
> [3] Worst-case Performance of Popular Approximate Nearest Neighbor Search Implementations: Guarantees and Limitations， NeurIPS, 2023.
>
> [4] Approximate Nearest Neighbor Search with Window Filters, ICML, 2024.
>
> [5] Efficient and robust approximate nearest neighbor search using hierarchical navigable small world graphs, TPAMI, 2018.

---

> ### Comment · Reviewer_osTr · 2024-11-28
> **Response**
>
> Thank you for responding to my comments.  I am convinced, especially after reading the other reviews, that my comment about the (lack of) novelty remains.  At this point, I will maintain my score and don't have further questions.

---

### Official Review · Reviewer_1FZ5 · 2024-11-04

**Soundness:** 3
**Presentation:** 2
**Contribution:** 3
**Rating:** 6
**Confidence:** 4

**Summary:**

This paper proposes ARDG as a new proximity graph index for maximum inner product search (MIPS). The observation is that only some vectors in the dataset can be the results of MIPS, which is called dominators. ARDG connects each vector to its nearby dominators and prunes the edges of the graph. Empirical results show that ARDG improves existing indexes for MIPS.

**Strengths:**

1.	The concept of dominators for MIPS is novel.

2.	Extensive theoretical analysis is conducted for the algorithm designs.

3.	The empirical results are good, showing large performance improvements over existing methods.

**Weaknesses:**

1.	The discussions of related work can be improved. In particular, [1] first uses IP-Voronoi cell, and it is natural that only vectors with a non-empty Voronoi cell can the results of MIPS (i.e., dominators in the paper). [2] is the first to observe that MIPS results cluster around large-norm vectors. [3] is a seminal work for LSH-based MIPS, and [4] is an important vector quantization method for MIPS.

[1] Non-metric similarity graphs for maximum inner product search
[2] Norm-ranging LSH for maximum inner product search.
[3] On Symmetric and Asymmetric LSHs for Inner Product Search
[4] Norm-Explicit Quantization: Improving Vector Quantization for Maximum Inner Product Search

2.	Presentation can be improved.
(1)	Algorithm 2 requires to connect each vector to its local dominators. What is the complexity of finding the dominators of a dataset?
(2)	The dominator ratio of the datasets (Table 2 in Appendix C) can be reported in Table 1of the paper.
(3)	Some index building time results (e.g., for the large datasets) can be included in the main paper.

3.	Experiments can be improved.
(1)	The implementation paragraph in Section 5.1 says that for each index, a unified parameter configuration is used for all datasets. This is not the common practice for evaluating vector search indexes as the index parameters (e.g., out-degree, ef-search) are usually tuned for each dataset. Please be carful to separate the tuning and search queries.
(2)	Section 4.1 first connects each vector to its local MIP neighbors and then conducts pruning. An ablation study for the pruning may be conducted.
(3)	The paper can report some results of searching other values of k (e.g., k=10, 20, 50). To save space, the main paper may only use the million-scale or larger datasets, while the results for the small datasets may be reported in the Appendix.

**Questions:**

How to identify the dominators for a vector dataset? What is the complexity of this step?

---

> ### Author Response · Authors · 2024-11-19
>
> We thank the reviewer for their valuable feedback. Below, we provide clarifications.
>
> **W1. The discussions of related work can be improved.**
>
> We have refined the related work based on the reviewer's comments, as detailed in Appendix A. We have added explanations for several renowned works and included more related works, which will help people better understand the relevant research fields.
>
> **W2. The presentation can be improved. (1) the complexity of finding the dominators. (2) move some results into the main paper.**
>
> **(1)** The complexity of selecting the real dominant nodes is $O(n^2)$, which is unacceptable in practical applications. In Section 4.1, we discuss the approach to get local dominators, which selects the two-hop neighbors of each node as the candidate set for its local dominators. Due to the favorable locality of our initial k-mip graph, it is an appropriate alternative. When the average degree of nodes is r, this can be efficiently computed in $O(r^2)$.
>
> **(2)** We have included the self-dominator ratio in Table 1 and relocated the index time and size for large-scale datasets in the main paper (Section 5.2).
>
> **W3. (1) Unified parameter configuration is used for all datasets. (2) Section 4.1 first connects each vector to its local MIP neighbors and then conducts pruning. An ablation study for the pruning may be conducted. (3) The paper can report some results of searching other values of k (e.g., k=10, 20, 50).**
>
> **(1)** Our choice of using uniform parameters was driven by practical considerations, acknowledging the diverse modalities and scales present in real-world data. Users often lack the experience to select optimal parameters for retrieval tasks, making the low-sensitivity algorithm beneficial. It ensures consistent and ideal retrieval results across various scenarios, thereby enhancing user experience [1,2].
>
> Furthermore, we have also fine-tuned the relevant parameters offline (to see the difference between various parameters). For graph algorithms, the most critical universal parameters are EFC (edge selection candidate count) and R (node degree), as they directly influence the index's quality. Empirically, a larger EFC generally improves performance by expanding the candidate node set, which aligns more closely with theoretical edge pruning. Conversely, R behaves as a convex function, with an optimal value that varies across datasets and it is difficult to select an optimal R. So, low-sensitivity algorithms will make the index run more robust in different scenarios.
>
> Query time is our foremost concern, to validate our approach, we further evaluated two representative graph baselines (ip-nsw and mobius graph) under different parameters (EFC-R) and datasets (ukbench, deep10m). The results showed that query speed variations with differing construction parameters were minimal (up to 6%), confirming the algorithms' low sensitivity and the robustness of our experimental setup. This low sensitivity ensures that our findings remain valid and applicable across a wide range of scenarios.
>
> |                          |            |            |            |            |
> | ------------------------ | ---------- | ---------- | ---------- | ---------- |
> | UKBench(recall@100=0.99) | **256-24** | **256-32** | **512-32** | **512-48** |
> | ip-NSW                   | 0.36       | 0.345      | 0.329      | 0.337      |
> | Mobius Graph             | 0.253      | 0.246      | 0.244      | 0.240      |
>
> | Deep10m (recall@100=0.98) | **256-24** | **256-32** | **512-32** | **512-48** |
> | ------------------------- | ---------- | ---------- | ---------- | ---------- |
> | ip-NSW                    | 0.561      | 0.548      | 0.543      | 0.529      |
> | Mobius Graph              | 0.737      | 0.728      | 0.722      | 0.695      |
>
> SCANN is sensitive to data scale and dimensionality. On the million-scale dataset, we used the official recommended parameters [3], while on the deep10M dataset, we conducted further fine-tuning. We have further elaborated on this point in the paper (Section 5.1), correcting previous ambiguities. We also look forward to further discussions on related issues.
>
> **(2)** The process of linking local MIP nodes is functionally equivalent to connecting edges to out-dominators, a component of our edge pruning rules. Given that our algorithm is a cohesive entity, we consider additional ablation experiments unnecessary.
>
> **(3)** We conducted experiments on the million-scale dataset with different top-k values (1,20,50,100) (Figure 7, Appendix D.3). The experiments demonstrate that our algorithm consistently exhibits stability across various top-k scenarios.
>
> [1] HVS: Hierarchical Graph Structure Based on Voronoi Diagrams for Solving Approximate Nearest Neighbor Search. VLDB, 2021.
>
> [2]FARGO: Fast Maximum Inner Product Search via Global Multi-Probing. VLDB, 2023.
>
> [3] https://github.com/google-research/google-research/tree/master/scann

---

> > ### Comment · Reviewer_1FZ5 · 2024-11-25
> > **After the author response**
> >
> > The author response addresses most of my concerns. The add exepriment shows that the baselines are robust to the parameter configurations. However, the question remains whether the proposed ARDG is robust to parameter configurations. Also, the experiment only covers a small range of paramter configurations (with R from 24 to 48), how about larger ranges (e.g., R=4, 8, 16, 32, 64)?

---

> > > ### Author Response · Authors · 2024-11-25
> > >
> > > Thank you for your response. We have conducted further experiments, demonstrating that graph-based algorithms remain robust across a broader range of parameters.  Notably, in large-scale datasets, it is not advisable to employ an excessively small $R$  as the node degree, as this can impair the graph's search capabilities.  The optimal value for $R$ is related to the intrinsic dimensionality of the dataset. Empirically, for a dataset of size $n$, we can select $R$ as $c*log_2(n)$, where $c \ge 1$ and $n$ represents the number of vectors.
> > >
> > > | UKBench(recall@100=0.99) | **256-8** | **256-16** | 256-24 | 512-32 | 512-48 | **512-64** |
> > > | ------------------------ | --------- | ---------- | ------ | ------ | ------ | ---------- |
> > > | ip-NSW                   | 0.380     | 0.368      | 0.36   | 0.329  | 0.337  | 0.341      |
> > > | Mobius Graph             | 0.271     | 0.255      | 0.253  | 0.244  | 0.240  | 0.242      |
> > > | ARDG                     | 0.259     | 0.221      | 0.212  | 0.192  | 0.20   | 0.213      |
> > >
> > > | **Deep10m (recall@100=0.98)** | **256-8** | **256-16** | 256-24 | 512-32 | 512-48 | **512-64** |
> > > | ----------------------------- | --------- | ---------- | ------ | ------ | ------ | ---------- |
> > > | ip-NSW                        | 1.09      | 0.596      | 0.561  | 0.543  | 0.529  | 0.538      |
> > > | Mobius Graph                  | 1.2       | 0.79       | 0.737  | 0.722  | 0.695  | 0.703      |
> > > | ARDG                          | 1.15      | 0.402      | 0.358  | 0.351  | 0.34   | 0.339      |

---

### Meta-Review · Area_Chair_r9EV · 2024-12-16

**Metareview:**

This paper introduces a novel approach to Maximum Inner Product Search (MIPS) by identifying and leveraging "dominators," which are data points that reside within their own Voronoi cells in the inner product space. This concept leads to the construction of the Approximate Relative Dominator Graph (ARDG), a graph index designed to efficiently prune the search space during MIPS queries.

The reviewers generally agree that the concept of dominators is novel and shows promise for MIPS.

However, reviewers also raise some important concerns:

- Limited Novelty: Some reviewers feel that the overall novelty of the work is somewhat incremental, building upon existing graph-based indexing techniques.
- Writing Quality: The clarity and presentation of the paper could be improved.
- Related Work Discussion: A more thorough and insightful discussion of related work is needed to better contextualize the contributions.
- Experimental Evaluation: While the experiments demonstrate performance gains, they are limited in scope. Expanding the evaluation to include more diverse datasets and comparisons with a wider range of MIPS techniques would strengthen the paper.

**Additional Comments On Reviewer Discussion:**

I went through the reviews, rebuttals and discussions. The discussion was nice and clarifications and improvements were discussed

---

### Decision · Program_Chairs · 2025-01-22

Reject